# Contrastive Sampling Chains in Diffusion Models

**Junyu Zhang**
Central South University
zhangjunyu@csu.edu.cn

**Daochang Liu**
The University of Sydney
daochang.liu@sydney.edu.au

**Shichao Zhang**
Central South University
zhangsc@csu.edu.cn

**Chang Xu**\*
The University of Sydney
c.xu@sydney.edu.au

## Abstract

The past few years have witnessed great success in the use of diffusion models (DMs) to generate high-fidelity images with the help of stochastic differential equations (SDEs). However, discretization error is an inevitable limitation when utilizing numerical solvers to solve SDEs. To address this limitation, we provide a theoretical analysis demonstrating that an appropriate combination of the contrastive loss and score matching serves as an upper bound of the KL divergence between the true data distribution and the model distribution. To obtain this bound, we utilize a contrastive loss to construct a contrastive sampling chain to fine-tuning the pre-trained DM. In this manner, our method reduces the discretization error and thus yields a smaller gap between the true data distribution and our model distribution. Moreover, the presented method can be applied to fine-tuning various pre-trained DMs, both with or without fast sampling algorithms, contributing to better sample quality or slightly faster sampling speeds. To validate the efficacy of our method, we conduct comprehensive experiments. For example, on CIFAR10, when applied to a pre-trained EDM, our method improves the FID from 2.04 to 1.88 with 35 neural function evaluations (NFEs), and reduces NFEs from 35 to 25 to achieve the same 2.04 FID. *The code is available at* [Contrastive-Sampling](#).

## 1 Introduction

Diffusion models (DMs) [57, 22] have emerged as powerful generative models, breaking records in image generation [67, 29, 26, 32], and finding rapid applications in other domains such as video generation [23], 3D point cloud generation [46], text-to-image generation [52], speech synthesis [7, 6], inverse problems [62, 30], and lossless compression [36]. While score-based generative models (SGMs) [57, 58] and denoising diffusion probabilistic models (DDPMs) [47, 22] are two branches of DMs, a certain parameterization reveals an equivalence between them [45]. In a seminal work, Song et al. [61] generalized DMs through the lens of stochastic differential equations (SDEs). Specifically, for any given stochastic diffusion process that progressively diffuses a data point into random noise with a continuum noise schedule, a DM learns to remove the added noise with a reverse-time SDE [1]. For SGMs, SDEs utilize deep neural networks to match the gradient of the log probability density with respect to data at each noise scale, which is dubbed denoising score matching [25, 68, 59]. In this manner, DMs enable not only exact likelihood computation [61] like variational auto-encoders (VAE) [37] but also higher sample quality [19] than widely-used generative adversarial networks (GANs) [3].

---

\*Corresponding author.

In practice, directly solving the reverse-time SDE to obtain an image is intractable. Instead, numerical solvers [69, 2, 44] are utilized to discretize the problem by constructing a discrete sampling chain with many sampling steps. Concretely, numerical solvers decompose the intractable integration of an SDE into multiple integration intervals [31] and solve them iteratively. However, due to the intractability of the integration in high dimensions, numerical solvers are unable to obtain the exact solution for the integration of each interval. Instead, they provide approximate solutions, which introduce a discretization error. This error represents the discrepancy between the approximate solution and the exact solution. It is important to note that any discretization scheme used by numerical solvers introduces discretization errors [42] in each integration interval. Consequently, there is a gap between the intermediate data distribution and its corresponding model distribution at each discretization step [75]. The accumulation of these discretization errors results in a larger gap between the true data distribution and its model distribution. It is worth noting that the estimation error, resulting from score matching or noise prediction during training, can also contribute to the gap between these two distributions [75, 17]. However, in this paper, we solely concentrate on the discretization error. From an integration point of view, the discretization error gradually decreases as the step size of the interval decreases. However, it cannot be completely avoided because the step size cannot be reduced to infinitesimal values.

To alleviate this issue, more discretization steps can be used to make each approximate solution of the discretization step closer to the exact solution, but this significantly increases the computational cost during sampling. For example, DDPM [22] requires 1000 steps to produce an image, with each step requiring the evaluation of the neural network once, which is substantially slower than GANs [28, 55]. While recent works have made it possible to achieve high-quality images with significantly reduced sampling steps [29, 32, 70], they still encounter the issue of discretization errors. On the other hand, some fast sampling DMs [27, 65, 72, 53] speed up sampling process for SDEs or ordinary differential equations (ODEs) [61], but suffer from a sharp decline in image quality since the discretization error grows larger. Hence, discretization errors have an severe impact on DMs.

More recently, several studies have been conducted to enhance numerical solvers which aim to reduce the discretization error. DPM-Solver [44] utilizes the semi-linear structure to avoid the corresponding discretization error by analytically computing the linear part of the solutions. By comparison, Liu et al. [42] combine high-order methods with Denoising Diffusion Implicit Model (DDIM) [56] to solve the ODE and achieve further acceleration. However, the accuracy of this approximation is not theoretically justified and may suffer from significant discretization error if the step size is large [75]. To fill the theoretical gap, Zhang and Chen [75] propose to utilize an exponential integrator to remedy the discretization error, achieving much better sample quality with smaller step numbers compared to previous fast sampling approaches. Orthogonal to this direction, our focus lies in minimizing the discretization error by optimizing the upper bound of the Kullback-Leibler (KL) divergence between the true sampling chain and a simulated chain at each time step.

The objective of our work is to establish a contrastive sampling chain to fine-tune any pre-trained DMs so as to reducing the discretization error, contributing to a small gap between data distribution and model distribution. Our method is motivated by the observation that minimizing the KL divergence between the true sampling chain and a simulated chain at each corresponding time step effectively reduces the discretization error. This reduction in discretization error directly translates into an improvement in sample quality. To achieve this, we propose minimizing a contrastive loss [11, 9, 21] that effectively reduces the gap between intermediate data distribution and the model data distribution at each time step. Specifically, by selecting instances on the sampling chain of a pre-trained DM for a same image as positive pairs, and choosing negative instances from other training images, a contrastive loss function is formed. To keep the pre-trained model stable in its generative ability, in practice, we combine the contrastive loss with the original generative loss using dynamic weighting schedules during the fine-tuning process. Moreover, the contrastive loss is optimized with backpropagation through time (BPTT) to spread the gradients on the whole sampling chain. In this manner, our method reduces the gap between true data distribution and model distribution, which improves sample quality without increasing sampling time overhead. Comprehensive experiments validate that our method can improve generation quality for various pre-trained models when using the same neural function evaluations (NFEs), or require less NFEs to achieve the same generation quality.

In a nutshell, our work makes the following contributions: 1) We demonstrate that the discretization error results in a gap between each intermediate data distribution and its corresponding model data distribution. 2) We analyze that an appropriate combination of contrastive loss and score

matching serves as an upper bound for the KL divergence between the data distribution and the model distribution. 3) We propose a contrastive sampling chain to fine-tune a pre-trained DM with the assistance of our derived upper bound. 4) We present dynamic weighting schedules and BPTT as optimization techiniques for our method.

## 2  Background

As mentioned previously, SGMs and DDPMs have been considered almost equivalent when a certain parameterization is applied. For the sake of simplicity, we solely focus on utilizing denoising score matching under SDEs for further investigation in this paper. Below, we present a comprehensive review of the entire DMs process with the lens of SGMs.

**Forward noising diffusion:** The forward diffusion of a DM for $D$-dimensional data is a linear diffusion described by the stochastic differential equation (SDE) [54]

$$dx = \boldsymbol{F}_t x dt + \boldsymbol{G}_t d\omega, \tag{1}$$

where $\boldsymbol{F}_t \in \mathbb{R}^{D \times D}$ denotes the linear drift coefficient, $\boldsymbol{G}_t \in \mathbb{R}^{D \times D}$ denotes the diffusion coefficient, and $\omega$ is a standard Wiener process. The diffusion Eq. (1) is initiated at the training data and simulated over a fixed time schedule $[0, T]$. Denote by $p_t(x_t)$ the marginal distribution of $x_t$ and by $p_{0t}(x_t \mid x_0)$ the conditional distribution from $x_0$ to $x_t$, then $p_0(x_0)$ represents the underlying distribution of the training data. The simulated stochastic process is represented by $\left\{x_t^{\text{SDE}}\right\}_{t \in [0,T]}$, where $p_T(x_T)$ is a prior $\pi(x_T)$ which is is easy to sample from, like Gaussian distribution. The parameters $\boldsymbol{F}_t$ and $\boldsymbol{G}_t$ are chosen such that the conditional marginal distribution $p_{0t}(x_t \mid x_0)$ is a simple Gaussian transition kernel, denoted as $\mathcal{N}(\mu_t x_0, \Sigma_t)$. Three popular SDEs in DMs are summarized by Song et al. [61], which are variance preserving SDE (VP SDE), variance exploding SDE (VE SDE) and sub-variance preserving (subVP SDE). Ideally, we enable to diffuse any data distribution to a prior distribution $\pi(x_T)$ with one of those three SDEs.

**Backward denoising diffusion:** Under mild assumptions [61], the forward diffusion Eq. (1) is associated with a reverse-time diffusion process

$$dx = \left[\boldsymbol{F}_t x - \boldsymbol{G}_t \boldsymbol{G}_t^T \nabla \log p_t(x)\right] dt + \boldsymbol{G}_t d\omega, \tag{2}$$

where $\omega$ denotes a standard Wiener process in the reverse-time direction, $\nabla \log p_t(x)$ denotes the gradient of the log probability density with respect to data at each time step $t$. In general, with a known prior distribution $\pi(x_T)$, one can model the data distribution $p_0(x_0)$ with Eq. (2) as $\boldsymbol{F}_t$ and $\boldsymbol{G}_t$ are fixed according to the forward SDEs. However, to solve Eq. (2), one needs to match the score function $\nabla \log p_t(x)$, which is not accessible.

**Training:** The basic idea of DMs is to use a time-dependent score matching network $s_\theta(x_t, t)$ to approximate the score $\nabla \log p_t(x)$. This is achieved by score matching techniques [25, 68, 59] where the score network $s_\theta$ is trained by minimizing the denoising score matching loss

$$\mathcal{J}_{\text{SM}}\left(\theta; \omega(t)\right) = \mathbb{E}_{t \sim U[0,T]} \mathbb{E}_{p(x_0)p_{0t}(x_t|x_0)} \left[\omega(t) \left\|\nabla \log p_{0t}(x_t|x_0) - s_\theta(x_t, t)\right\|_2^2\right]. \tag{3}$$

Here $\nabla \log p_{0t}(x_t|x_0)$ has a closed form expression as $p_{0t}(x_t|x_0)$ is a simple Gaussian distribution which represents the discretized form of a given SDE, $\omega(t)$ denotes a time-dependent weighting function. This loss can be evaluated using empirical samples by Monte Carlo methods and thus standard stochastic optimization algorithms can be used for training.

**Sampling:** Once the score network $s_\theta(x_t, t) \approx \nabla \log p_t(x)$ is matched for almost all $x \in \mathbb{R}^D$ and $t \sim U[0, T]$, it can be used to generate new samples by solving the backward SDE Eq. (2) with $\nabla \log p_t(x)$ replaced by $s_\theta(x_t, t)$. It turns out there are infinitely many diffusion processes one can use. In this work, to show the scalability of our method, we consider a general expression of SDEs

$$dx^{\text{SDE}} = \left[\boldsymbol{F}_t x^{\text{SDE}} - \frac{1 + \lambda^2}{2} \boldsymbol{G}_t \boldsymbol{G}_t^T s_\theta(x^{\text{SDE}}, t)\right] dt + \lambda \boldsymbol{G}_t d\omega, \tag{4}$$

parameterized by $\lambda > 0$. When $\lambda = 0$, Eq. (4) reduces to an ordinary differential equation (ODE) known as the probability flow ODE [8]. The reverse-time diffusion Eq. (2) with an approximated score is a special case of Eq. (4) with $\lambda = 1$.

# 3 Discretization Error Analysis

To generate a new image, one can sample $x_T$ from a standard distribution $\pi(x_T)$ and solve Eq. (4) to obtain an image $x_0^{\text{SDE}}$. However, in practice, exact solutions are not attainable as it is intractable to solve Eq. (4) directly. To remedy this, one needs to discretize Eq. (4) over time to get an approximated solution, which leads to a discretization error. For brevity, we next investigate the discretization error of solving the probability flow ODE ($\lambda = 0$)

$$\frac{dx}{dt} = \boldsymbol{F}_t x - \frac{1}{2}\boldsymbol{G}_t\boldsymbol{G}_t^T s_\theta(x_t^{\text{SDE}}, t), \tag{5}$$

and $x_t^{\text{SDE}}$ represents the discretization samples solved by Eq. (4) with a numerical solver. The exact solution to this ODE is

$$x_t^{\text{SDE}} = \Psi(t, T)x_T^{\text{SDE}} + \int_T^t \Psi(t, \tau)\left[-\frac{1}{2}\boldsymbol{G}_\tau\boldsymbol{G}_\tau^T s_\theta\left(x_\tau^{\text{SDE}}, \tau\right)\right]d\tau, \tag{6}$$

where $\Psi(t, T)$ satisfying $\frac{\partial \Psi(t,T)}{\partial(t)} = F_T\Psi(t, T)$, $\Psi(t, T) = I$ is known as the transition matrix from time $T$ to $t$ associated with $F_\tau$. There exist many numerical solvers for Eq. (5) associated with different discretization schemes to approximate Eq. (6). As the discretization step size goes to infinitesimal, the solutions obtained from all these methods converge to that of Eq. (5).

However, the performances of these methods can be dramatically different when the step size is large. On the other hand, to achieve fast sampling in DMs, one needs to approximately solve Eq. (5) with a small number of discretization steps, and thus large step size. Concretely, the discretization of Eq. (4) equals to build a discretization sampling chain, which is $\left\{x_t^{\text{SDE}}\right\}_{t\in[0,T]}$, and iteratively convert it to a new image $x_0^{\text{SDE}}$ with randomly initialize a sample from $x_T$, where $T$ is the total sampling steps. Following the philosophy of discretization sampling, a small number of discretization steps is equivalent to small sampling steps $T$ and a large step size $\Delta t$ from $x_T$ to $x_t$. When Euler method applied to Eq. (5), the discretization form can be expressed as

$$x_t^{\text{SDE}} = x_T^{\text{SDE}} - \left[\boldsymbol{F}_T x_T^{\text{SDE}} - \frac{1}{2}\boldsymbol{G}_T\boldsymbol{G}_T^T s_\theta(x_T^{\text{SDE}}, T)\right]\Delta t. \tag{7}$$

From the integral point of view, samples $x_0^{\text{SDE}}$ obtained from Eq. (7) equals the exact solutions of Eq. (5) if and only if the step size between $x_T^{\text{SDE}}$ and $x_t^{\text{SDE}}$ goes to infinitesimal. In practice, it's impossible to achieve it, especially the fast sampling demand in DMs that requires small $T$ even one-step sampling [63], which means $x_0^{\text{SDE}}$ is an approximate solution. Hence, the discretization error is essentially a gap between the approximate solution and the exact solution

$$\begin{aligned}\Delta(x_t^{\text{SDE}}, x_T^{\text{SDE}}) = \frac{1}{2}\left\|\int_T^t \Psi(t, \tau)\left[\boldsymbol{G}_\tau\boldsymbol{G}_\tau^T s_\theta\left(x_\tau^{\text{SDE}}, \tau\right)\right]d\tau\right. \\ \left. - \left[2 * \boldsymbol{F}_T x_T^{\text{SDE}} - \boldsymbol{G}_T\boldsymbol{G}_T^T s_\theta(x_T^{\text{SDE}}, T)\right]\Delta t\right\|_2^2.\end{aligned} \tag{8}$$

In light of this, the presence of $\Delta(x_t^{\text{SDE}}, x_T^{\text{SDE}})$ creates a gap between $p_t(x_t)$ and $p_t^{\text{SDE}}(x_t^{\text{SDE}})$ since each sample $x_t^{\text{SDE}}$ only approximates the exact solution $x_t$. Consequently, one enables to optimize the KL divergence between $p_t(x_t)$ and $p_t^{\text{SDE}}(x_t^{\text{SDE}})$ to minimize this gap, denoted as $D_{\text{KL}}\left(p_t\|p_t^{\text{SDE}}\right)$. By minimizing this KL term, one can effectively reduce the discretization error. This is because each $p_t^{\text{SDE}}(x_t^{\text{SDE}})$ from the simulated chain will closely approximate $p_t(x_t)$ from the true sampling chain.

# 4 Theoretical Analysis

It is well-known that maximizing the log-likelihood of a probabilistic model is equivalent to minimizing the KL divergence from the data distribution to the model distribution [60, 43]. Similarly, in order to improve the log-likelihood of DMs, we can optimize the KL divergence $D_{\text{KL}}\left(p_0\|p_0^{\text{SDE}}\right)$ between the true distribution $p_0(x_0)$ and its corresponding model distribution $p_0^{\text{SDE}}(x_0^{\text{SDE}})$.

In what follows, we demonstrate that with a appropriate combination of $D_{\text{KL}}\left(p_t\|p_t^{\text{SDE}}\right)$ and score matching losses $\mathcal{J}_{\text{SM}}\left(\theta; g(t)^2\right)$, they actually becomes an upper bound on $D_{\text{KL}}\left(p_0\|p_0^{\text{SDE}}\right)$ [60]. Notably, the weighting function $\omega(t)$ in Eq. (3) is replaced by $g(t)^2$ which is the diffusion coefficient of a SDE in Eq. (1).

**Theorem 1.** *Let $p_0(x_0)$ be the true data distribution, $\pi(x_T)$ be a known prior distribution. Suppose $\{x_t\}_{t\in[0,T]}$ is a stochastic process defined by the SDE in Eq. (1) with $x_0 \sim p_0(x_0)$, where the marginal distribution of $x_t$ is denoted as $p_t(x_t)$. By comparison, $\{x_t^{\mathrm{SDE}}\}_{t\in[0,T]}$ is another stochastic process obtained by the reverse-SDE in Eq. (4) from a pre-trained DM with $x_T^{\mathrm{SDE}} \sim p_T(x_T)$ and $x_0^{\mathrm{SDE}} \sim p_0^{\mathrm{SDE}}(x_0^{\mathrm{SDE}})$, where the marginal distribution of $x_t^{\mathrm{SDE}}$ is denoted as $p_t^{\mathrm{SDE}}(x_t^{\mathrm{SDE}})$. Under some regularity conditions detailed in Appendix B, we have*

$$D_{\mathrm{KL}}\left(p_0 \| p_0^{\mathrm{SDE}}\right) \leq D_{\mathrm{KL}}\left(p_t \| p_t^{\mathrm{SDE}}\right) + \mathcal{J}_{\mathrm{SM}}\left(\theta; g(t)^2\right). \tag{9}$$

*Sketch of proof.* Let $\boldsymbol{\mu}$ and $\boldsymbol{\nu}$ denote the path measures [40, 38] of SDEs in Eq. (1) and Eq. (4) with $\lambda = 1$. Conceptually, $\boldsymbol{\mu}$ is the joint distribution of the forward diffusion process $\{x_t\}_{t\in[0,T]}$ given by Eq. (1) and $\boldsymbol{\nu}$ represents another joint distribution of the process $x_0^{\mathrm{SDE}} \sim p_0^{\mathrm{SDE}}(x_0^{\mathrm{SDE}})$ simulated by Eq. (4). Since we can marginalize $\boldsymbol{\mu}$ and $\boldsymbol{\nu}$ to obtain distributions $p_0(x_0)$ and $p_0^{\mathrm{SDE}}(x_0^{\mathrm{SDE}})$, the data processing inequality [60] gives $D_{\mathrm{KL}}\left(p_0 \| p_0^{\mathrm{SDE}}\right) \leq D_{KL}(\boldsymbol{\mu}_t \| \boldsymbol{\nu}_t)$, where $\boldsymbol{\mu}_t$ and $\boldsymbol{\nu}_t$ are subset of $\boldsymbol{\mu}$ and $\boldsymbol{\nu}$ respectively. Intuitively, $D_{\mathrm{KL}}\left(p_0 \| p_0^{\mathrm{SDE}}\right)$ is a subset of the path measure between $\boldsymbol{\mu}$ and $\boldsymbol{\nu}$ which accumulate all the KL divergence $D_{\mathrm{KL}}\left(p_t \| p_t^{\mathrm{SDE}}\right)_{t\in[0,T]}$ [40]. Hence, $D_{\mathrm{KL}}\left(p_0 \| p_0^{\mathrm{SDE}}\right) \leq D_{KL}(\boldsymbol{\mu}_t \| \boldsymbol{\nu}_t) = \sum_i^t D_{\mathrm{KL}}\left(p_i \| p_i^{\mathrm{SDE}}\right), t \sim [0,T]$. From the chain rule for the KL divergence [40], we also have $D_{KL}(\boldsymbol{\mu}_t \| \boldsymbol{\nu}_t) = D_{\mathrm{KL}}\left(p_t \| p_t^{\mathrm{SDE}}\right) + \mathbb{E}_{p_i(\mathbf{z})}\left[D_{\mathrm{KL}}\left(\boldsymbol{\mu}(\cdot \mid x_i = \mathbf{z}) \| \boldsymbol{\nu}\left(\cdot \mid x_i^{\mathrm{SDE}} = \mathbf{z}\right)\right)\right]$ with $i \in [0,t]$, where the KL divergence in the final term can be computed by applying the Girsanov theorem [48] to Eq. (4) and the reverse-time SDE of Eq. (1). Combining above analysis completes the proof, detailed see in Appendix B.

Though we demonstrate that the combination of score matching losses $\mathcal{J}_{\mathrm{SM}}\left(\theta; g(\cdot)^2\right)$ and a KL term $D_{\mathrm{KL}}\left(p_t \| p_t^{\mathrm{SDE}}\right)$ is an upper bound of $D_{\mathrm{KL}}\left(p_0 \| p_0^{\mathrm{SDE}}\right)$, it is intractable to optimize Eq. (9) due to the unknown function form of $p_t$. To circumvent this problem, we combined with the mutual information (MI) theory [50] that MI between $p_t(x_t)$ and $p_t^{\mathrm{SDE}}(x_t^{\mathrm{SDE}})$ can be expressed as $I(p_t^{\mathrm{SDE}}(x_t^{\mathrm{SDE}}), p_t(x_t)) \leq D_{KL}(p_t^{\mathrm{SDE}} \| p_t)$. Moreover, when applied the Jensen's inequality to this term, we enable to obtain an upper bound of $D_{KL}(p_t \| p_t^{\mathrm{SDE}})$ by a InfoNCE loss: $I_{\mathrm{InfoNCE}}(x_t^{\mathrm{SDE}}, x_j^{\mathrm{SDE}}) \geq D_{KL}(p_t \| p_t^{\mathrm{SDE}})$ [49], where $j \in [0,T]$ and $j \neq t$ (detailed in Appendix B). Hence, we obtain a new upper bound of the KL divergence of $D_{\mathrm{KL}}\left(p_0 \| p_0^{\mathrm{SDE}}\right)$ when applies this term to Eq. (9), which is

$$D_{\mathrm{KL}}\left(p_0 \| p_0^{\mathrm{SDE}}\right) \leq \Gamma(x_j^{\mathrm{SDE}}, x_t^{\mathrm{SDE}}; \beta(t)) + \mathcal{J}_{\mathrm{SM}}\left(\theta; g(t)^2\right), \tag{10}$$

where $\Gamma(x_t^{\mathrm{SDE}}, x_j^{\mathrm{SDE}}; \beta(t)) = \beta(t) * I_{\mathrm{InfoNCE}}(x_t^{\mathrm{SDE}}, x_j^{\mathrm{SDE}})$ is a scaling of $I_{\mathrm{InfoNCE}}$ with the weighting function $\beta(t)$. Hence, it is obvious that the the combination of score matching losses $\mathcal{J}_{\mathrm{SM}}$ and a contrastive loss $I_{\mathrm{InfoNCE}}(x_t^{\mathrm{SDE}}, x_j^{\mathrm{SDE}})$ in Eq. (10) is an upper bound of the KL divergence between data distribution $p_0(x_0)$ and the model distribution $p_0^{\mathrm{SDE}}(x_0^{\mathrm{SDE}})$.

Importantly, the score matching term $\mathcal{J}_{\mathrm{SM}}\left(\theta; g(t)^2\right)$ is stable and almost fixed in the Eq. (10), as we fine-tune the pre-trained DMs that the score network $s_\theta(x_t) \approx \nabla \log p_t(x)$ is matched for almost all $x \in \mathbb{R}^D$ and $t \sim U[0,T]$. In this context, optimizing the Eq. (10) equals to minimize $\Gamma(x_t^{\mathrm{SDE}}, x_j^{\mathrm{SDE}}; \beta(t))$, which is the upper bound of $D_{\mathrm{KL}}\left(p_t \| p_t^{\mathrm{SDE}}\right)$ [50]. Therefore, minimizing $D_{\mathrm{KL}}\left(p_t \| p_t^{\mathrm{SDE}}\right)$ results in a smaller gap between $x_t$ and $x_t^{\mathrm{SDE}}$, indicating that the approximate solution obtained through numerical solvers is closer to the exact solution with the help of contrastive loss. On the other hand, each minimized $D_{\mathrm{KL}}\left(p_t \| p_t^{\mathrm{SDE}}\right)$ will decreased the corresponding $\Delta(x_t^{\mathrm{SDE}}, x_T^{\mathrm{SDE}})$, contributing to a smaller discretization error between step $T$ and any step $t$. From this perspective, utilizing a contrastive loss to increase the similarity between $x_j^{\mathrm{SDE}}$ and each $x_t^{\mathrm{SDE}}$ in a sampling chain is actually minimizing $D_{\mathrm{KL}}\left(p_t \| p_t^{\mathrm{SDE}}\right)_{t\in[0,T]}$. Hence, by minimizing the Eq. (10), one can achieve a better model distribution $p_0^{\mathrm{SDE}}(x_0^{\mathrm{SDE}})$ with a reduction in all the discretization error present in the sampling chain. Moreover, the decreased discretization error also benefits the faster sampling speed [44, 2, 75, 63].

## 5   Methodology

Contrastive learning has recently achieved remarkable performance [4, 12, 21] and has made significant waves in deep learning for computer vision tasks [35, 51]. These influential works leverage

the contrastive loss to bring similar images closer together in high-dimensional space, resulting in notable improvements in downstream tasks. Building on the insights provided in Section 4, we employ the InfoNCE loss [49] to formulate our objective function. By doing so, we effectively reduce the discretization error between $x_t^{\text{SDE}}$ and $x_{t-1}^{\text{SDE}}$ by optimizing the upper bound of $D_{\text{KL}}\left(p_t \| p_t^{\text{SDE}}\right)$. Consequently, numerical solvers are capable of providing more precise solutions when solving Eq. (5), and a fine-tuned DM naturally enhances the quality of generated samples. In the following sections, we will present our method, focusing on constructing contrastive sampling chains and optimizing DMs.

## 5.1 Contrastive Sampling Chain

As previously mentioned, our objective is to enhance the quality of samples by reducing the discretization error. Additionally, we illustrate that the presence of this discretization error creates a gap between $x_t^{\text{SDE}}$ and $x_t$. Analogously, optimizing this gap is equivalent to minimizing the corresponding discretization error. While directly decreasing this gap is infeasible, minimizing its upper bound provides an alternative approach to achieve the same outcome. Leveraging Theorem 1 and Eq. (10), we propose a contrastive sampling chain for fine-tuning pre-trained DMs using the contrastive loss. To accomplish this, we construct the contrastive loss and combine it with the score matching loss $\mathcal{J}_{\text{SM}}\left(\theta; g(t)^2\right)$ to jointly update the parameter $\theta$.

To construct the contrastive loss, we randomly select an image $x_t^{\text{SDE}}$ and another image $x_j^{\text{SDE}}$ from the defined sampling chain to form a positive pair. Meanwhile, negative instances are sampled from the training images. Next, we extract 128-dimensional latent representations of these images using the pre-trained MoCov2 [11]. The contrastive loss, known as the InfoNCE loss [49], is then computed in the subsequent steps. When applying the pre-trained MoCov2 $\boldsymbol{E}$, the InfoNCE loss can be expressed as follows:

$$I_{\text{InfoNCE}}(x_t^{\text{SDE}}, x_j^{\text{SDE}}) = \log \frac{\exp\left(\boldsymbol{E}(x_t^{\text{SDE}}) \cdot \boldsymbol{E}(x_j^{\text{SDE}})/\tau\right)}{\exp\left(\boldsymbol{E}(x_j^{\text{SDE}}) \cdot \boldsymbol{E}(x_t^{\text{SDE}})/\tau\right) + \sum_{k^-} \exp\left(\boldsymbol{E}(x_j^{\text{SDE}}) \cdot \boldsymbol{E}(x^-)/\tau\right)}, \quad (11)$$

where $x_j^{\text{SDE}}$ and $x_t^{\text{SDE}}$ form a positive pair which all generated via iteratively calculating Eq. (7) during the fine-tuing process. By comparsion, $x^-$ are negative instances which sampled from training images and $\tau$ is a temperature hyper-parameter. Conceptually, the InfoNCE loss is crafted to bring similar features closer, thereby reducing the distance between $x_t^{\text{SDE}}$ and $x_j^{\text{SDE}}$. In this manner, the sampling chain will become tighter and the discretization error decreased accordingly because the integration interval between sampling steps decreased.

Though we show that $I_{\text{InfoNCE}}(x_t^{\text{SDE}}, x_j^{\text{SDE}})$ is an upper bound of $D_{KL}(p_t \| p_t^{\text{SDE}})$ in Section 4, directly using Eq. (11) to fine-tune DMs will destroy the optimal result of the previous score matching in practice. To circumvent this problem, we combine the generative loss Eq. (3) and contrastive loss Eq. (11) to optimize pre-trained DMs

$$\mathcal{L}_\theta = \mathcal{J}_{\text{SM}}\left(\theta; g(t)^2\right) + \beta(t) * I_{\text{InfoNCE}}(x_t^{\text{SDE}}, x_j^{\text{SDE}}), \quad (12)$$

where $\beta(t)$ is the weighting schedule to balance score matching term and contrastive loss term, which will be analyzed in detail in the following subsection. In Eq. (12), we apply the contrastive loss via calculating $x_t^{\text{SDE}}$ and all $x_j^{\text{SDE}}$ in the same sampling chain with Eq. (11). Though we amplify the $I_{\text{InfoNCE}}$ with $\beta(t)$, the combination of score matching term $\mathcal{J}_{\text{SM}}\left(\theta; g(t)^2\right)$ is still an upper bound of $D_{\text{KL}}\left(p_0 \| p_0^{\text{SDE}}\right)$.

## 5.2 Optimization

Considering the previous discussion, it is crucial to strike a balance between the InfoNCE loss and the generative loss. Placing excessive emphasis on the InfoNCE loss may disrupt the stability of the pre-trained generative task, while assigning a high weight to the generative loss may have a negligible impact. To address this, we draw inspiration from the fact that the InfoNCE loss quantifies the similarity between $x_t^{\text{SDE}}$ and $x_j^{\text{SDE}}$, where the similarity increases as $x_j^{\text{SDE}}$ approaches $x_t^{\text{SDE}}$. Consequently, it is reasonable to implement a dynamic weighting schedule based on the time step $t$. This schedule assigns higher weights to the loss when $x_j^{\text{SDE}}$ is closer to $x_t^{\text{SDE}}$, and lower weights otherwise. To strike an appropriate balance between the two losses during the refinement of pre-trained DMs, we have devised two dynamic weighting schedules: the linear and nonlinear weighting schedules.

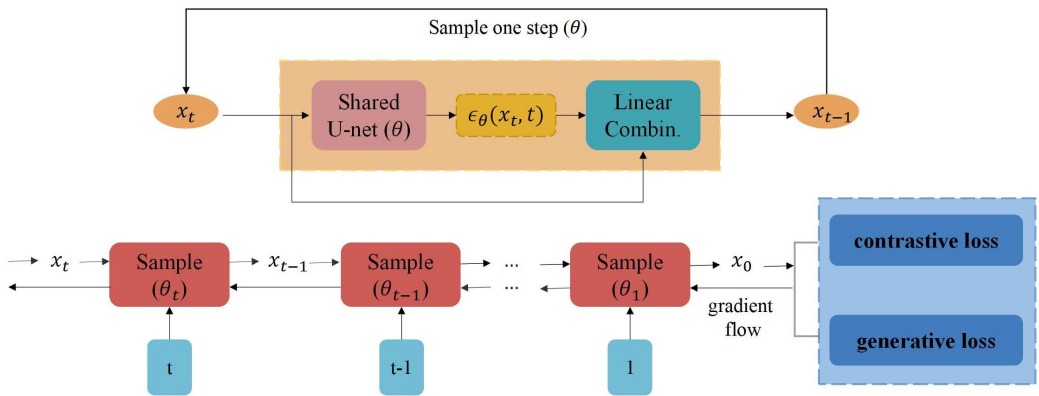

Figure 1: Optimizing Diffusion Models via Back Propogation Through Time.

**Linear weighting schedule:** The weights in this schedule increase progressively with the discretization step as it gets closer to $x_j^{\text{SDE}}$. For instance, larger weights are assigned to $x_t^{\text{SDE}}$ if it shares a higher similarity with $x_j^{\text{SDE}}$. In contrast, smaller weights are assigned to the earlier step of the sampling chain. At this extreme, a linear weighting schedule can be expressed as

$$\beta(t) = \alpha * (T - t), \tag{13}$$

where $\alpha$ is a hyper-parameter to scale the weighting schedule.

**Nonlinear weighting schedule:** The nonlinear weighting schedule works in a similar fashion to the linear weighting schedule, with the only difference being that the weights increase at different rates. Specifically, we calculate the noise ratio in $x_t^{\text{SDE}}$ compared to $x_j^{\text{SDE}}$ using the peak signal-to-noise ratio (PSNR) [76, 14] and subsequently apply the PSNR as the elements of this weighting schedule. Therefore, the nonlinear weighting schedule can be expressed as:

$$\beta(t) = \alpha * \text{PSNR}(x_j^{\text{SDE}}, x_t^{\text{SDE}}). \tag{14}$$

In general, the nonlinear weighting schedule is more consistent with the principle of the sampling chain in DMs, as the weights gradually remove the noise in the samples. However, in practice, both weighting schedules demonstrate similar effectiveness.

**Computational graph:** As mentioned earlier, we construct a contrastive sampling chain to fine-tune a pre-trained DM. Though we utilize Eq. (6) to directly obtain the sample $x_t^{\text{SDE}}$, in practice, $x_t^{\text{SDE}}$ is iterately solved by Eq. (4). Hence, we need to consider the accumulated discretization error such as $\Delta(x_{T-1}^{\text{SDE}}, x_T^{\text{SDE}})$ and $\Delta(x_{T-2}^{\text{SDE}}, x_{T-1}^{\text{SDE}})$, leads to a larger gap between $x_t^{\text{SDE}}$ and $x_T^{\text{SDE}}$. To address this issue and influence the parameters of previous steps while optimizing the KL divergence $D_{\text{KL}}\left(p_t \| p_t^{\text{SDE}}\right)$ for the current step $t$, we propose to fine-tune pre-trained DMs using a gradient propagation mechanism similar to BPTT [73], shown in Figure 1.

Specifically, for a given sampling chain, we randomly select one sample $x_t^{\text{SDE}}$ and another sample $x_j^{\text{SDE}}$, and calculate the InfoNCE loss with the previously mentioned weighting function. By combining it with the generative loss (Eq. (3)), we employ an optimizer to propagate the gradients of these two losses in reverse order through the chain from the current step $t$ to the final step $T$. In this manner, the gradients influence the entire sampling chain and update the parameters of DMs at each time step $t$. Consequently, the training objective can be reformulated as follows:

$$\mathcal{L}_\theta = \mathcal{J}_{\text{SM}}\left(\theta; \omega(t)\right) + \Gamma(x_t^{\text{SDE}}, x_j^{\text{SDE}}; \beta(t)); t \sim U[0, T], \tag{15}$$

where $\Gamma(x_t^{\text{SDE}}, x_j^{\text{SDE}}; \beta(t))$ represents the InfoNCE loss Eq. (11) with weighting function $\beta(t)$. By following this design philosophy, the InfoNCE loss does not solely update parameters independently; rather, it operates as a unified entity. Consequently, we enable to optimize the current KL divergence $D_{\text{KL}}\left(p_t \| p_t^{\text{SDE}}\right)$ and simultaneously influence KL divergences in previous time steps. This approach ensures a cohesive and synchronized optimization process throughout the entire contrastive sampling chain. In summary, our method effectively mitigates the cumulative discretization errors, as each discretization error is correspondingly reduced. Below, we demonstrate experiment results to further prove our analysis.

Table 1: Performance on CIFAR-10. Our method, denoted as C++, has better quality than baselines with the same NFEs, and fewer NFEs for the same quality.

| Method | Space | NFE↓ | NLL↓ | FID↓ |
|---|---|---|---|---|
| *Unconditional* | | | | |
| VDM [36] | Data | 1000 | **2.49** | 7.41 |
| DDPM [22] | Data | 1000 | 3.75 | 3.17 |
| iDDPM [47] | Data | 1000 | 3.37 | 2.90 |
| STDDPM [34] | Data | 2000 | 2.91 | 2.47 |
| INDM [33] | Latent | 2000 | 3.09 | 2.28 |
| CLD-SGM [20] | Data | 312 | 3.31 | 2.25 |
| NCSN++ [61] | Data | 2000 | 3.45 | 2.20 |
| LSGM [67] | Latent | 138 | 3.43 | 2.10 |
| LSGM-C++ (Ours) | Latent | **100** | 3.40 | 2.10 |
| LSGM-C++ (Ours) | Latent | 138 | **3.40** | **1.99** |
| EDM [29] | Data | 35 | 2.60 | 2.04 |
| EDM-C++ (Ours) | Data | **25** | 2.55 | 2.04 |
| EDM-C++ (Ours) | Data | 35 | **2.55** | **1.88** |
| *Conditional* | | | | |
| NCSN++-G [5] | Data | 2000 | - | 2.25 |
| EDM | Data | 35 | 2.60 | 1.82 |
| EDM-C++ (Ours) | Data | **27** | 2.55 | 1.82 |
| EDM-C++ (Ours) | Data | 35 | **2.55** | **1.73** |

Table 2: Performance on fast samplers on CIFAR-10 and ImageNet 64x64 with FID reported. DDIM [56], DPM-Solver [44], and DEIS [75] are classical training-free fast samplers. "†" means the actual NFE is smaller than the NFE [44] given in the table. "-" represents that the FID for this NFE is not shown in original papers.

| NFE / Method | 10 | 12 | 15 | 20 | 50 |
|---|---|---|---|---|---|
| *DDPM (CIFAR-10) [22]* | | | | | |
| DDIM [56] | 13.36 | - | - | 6.84 | 4.67 |
| DDIM-C++ (Ours) | **11.52** | - | - | **6.09** | **4.12** |
| DPM-Solver-3 [44] | †24.37 | 8.20 | †5.73 | †5.43 | †5.29 |
| DPM-Solver-3-C++ (Ours) | †**21.13** | **7.14** | **5.22** | †**5.06** | †**4.91** |
| *SDE VP (CIFAR-10) [61]* | | | | | |
| DPM-Solver-3 | †54.56 | 6.03 | 3.55 | †2.90 | †2.65 |
| DPM-Solver-3-C++ (Ours) | †**43.81** | **5.41** | **3.25** | †**2.84** | †**2.60** |
| DEIS-tAB3 [75] | 4.17 | - | 3.37 | 2.86 | 2.57 |
| DEIS-tAB3-C++ (Ours) | **4.02** | - | **3.20** | **2.77** | **2.48** |
| *IDDPM (ImageNet) [47]* | | | | | |
| DPM-Solver-3 | †57.48 | 24.62 | 19.76 | †18.95 | †17.52 |
| DPM-Solver-3-C++ (Ours) | †**50.63** | **22.65** | **19.45** | †**18.73** | †**17.49** |
| *IDDPM (ImageNet Conditional) [47]* | | | | | |
| DEIS-tAB3 | 6.65 | 3.99 | †3.67 | 3.10 | 2.69 |
| DEIS-tAB3-C++ (Ours) | **6.59** | **3.91** | †**3.60** | **3.07** | **2.67** |
| DPM-Solver-2 [44] | 7.93 | 5.36 | †4.46 | 3.42 | 2.82 |
| DPM-Solver-2-C++ (Ours) | **7.78** | **5.22** | †**4.38** | **3.36** | **2.80** |

# 6 Experiments

In this section, we demonstrate effectiveness of the contrastive sampling chain by experimental results, namely higher image quality, better log-likelihood, or slightly faster sampling speed. Comprehensive experiments are conducted on various datasets, including CIFAR-10, CelebA/FFHQ 64x64, and ImageNet 64x64. We first utilize contrastive sampling chain to improve pre-trained DMs, for which we select EDM [29] and LSGM [67] on CIFAR-10, EDM [29] on FFHQ, and STDDPM [34] on CelebA. Additionally, we further verify the performance of our method with sampling chains defined by training-free fast sampling methods, such as DDIM [56], DPM-Solver [44], and DEIS [75], when they are combined with various pre-trained DMs. Specifically, we combine these fast sampling methods with IDDPM [47] on ImageNet 64x64, DDPM [22] and SDE VP [61] on CIFAR-10. Our objective is not to focus on achieving state-of-the-art metrics in generative models, but rather to demonstrate the significant performance of our method in enhancing pre-trained DMs, either combined with or without training-free fast-sampling methods.

It is worth noting that we maintain all the training settings of the pre-trained DMs and only modify the part that constructs the contrastive loss, thereby ensuring fair comparison and demonstrating the flexibility of our method. Similarly, to showcase the applicability of our method to sampling chains defined by fast sampling methods, we replace the original chains provided by pre-trained DMs with new chains for fast sampling, while retaining all other settings the same as fine-tuning pre-trained DMs. Once the fine-tuning process is completed, we test the performance of the refined model by drawing 50,000 samples from it and measuring the widely adopted Fréchet Inception Distance (FID) score, Negative Log-Likelihoods (NLL), and Neural Function Evaluations (NFEs), where lower values indicate better performance. Moreover, we also present the generated images for qualitative comparison, shown in Figure 2.

## 6.1 Performance on Pre-trained DMs

We first showcase the performance of our method in refining the original pre-trained DMs. Tables 1, 4 and 5 present the results of our method on CIFAR-10, FFHQ and CelebA respectively. Our method can achieve better generation quality than baselines when using the same NFEs. On the other hand, our method requires less NFEs to achieve the same quality as baselines. To evaluate on CIFAR-10, we apply our method to LSGM [67] and EDM [29] under unconditional or conditional settings. For a fair comparison, we report performances on EDM [29] under the random seed from [32, 63]. Specifically, our method improves LSGM from 2.10 FID to 1.99 FID and achieves slightly better NLL with 138 NFEs, while requiring only 100 NFEs to attain the same 2.10 FID. Moreover, we enhance EDM from 2.04 FID to 1.88 FID and reduce the NLL from 2.60 to 2.55 with 35 NFEs, while requiring only 25 NFEs for the same 2.04 FID. Furthermore, we improve the performance on conditional EDM, with 1.82 FID reduced to 1.73 FID using 35 NFEs or maintaining the same 1.82 FID with only 27 NFEs. For the evaluation on CelebA and FFHQ, we apply our method to refine STDDPM [34] and EDM

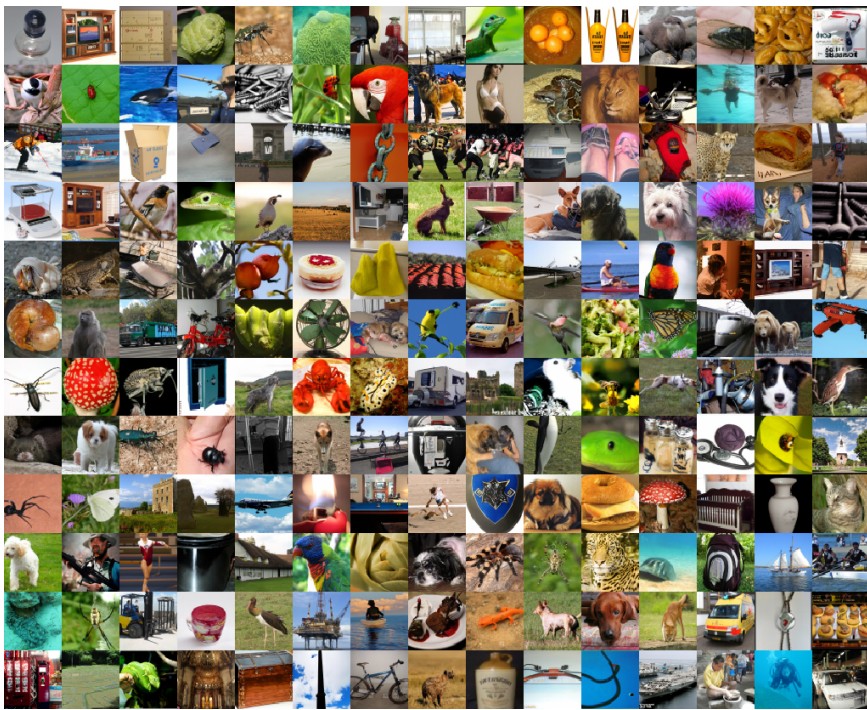

Figure 2: Randomly selected ImageNet Samples from IDDPM Improved by Our Method.

[29] respectively. Specifically, our method improves the performance of STDDPM on CelebA 64x64 from 1.90 FID to 1.73 FID with the same 131 NFEs, and achieves the same 1.90 FID with fewer NFEs (100). Similarly, we significantly enhance the performance of EDM on FFHQ 64x64 from 2.39 FID to 2.07 FID with the same 79 NFEs, and achieve the same 2.39 FID with fewer NFEs (63).

## 6.2 Performance on Fast Samplers

To demonstrate the applicability of our approach to sampling chains defined by training-free fast sampling methods, we carry out a sequence of experiments on CIFAR-10 and ImageNet 64x64, with detailed results in Table 2. We test our method on different fast samplers when combined with various pre-trained DMs. We utilize DDIM [56], DPM-Solver [44], and DEIS [75] to replace the original chains given by DDPM [22], SDE VP [61] and IDDPM [47]. Subsequently, for a pre-trained model with sampling chain defined by these fast samplers, we refine it with Eq. (15). In this manner, our method remarkably improves the results as illustrated in Table 2. For example, our method obtains better sample quality when applied to refine the SDE VP with DEIS. Concretely, our method improve the FID from 4.17 to 4.02 in 10 NFEs and increase the 2.57 FID to 2.48 FID in 50 NFEs. In comparison, the improvement on ImageNet is not significant as our main objective is to demonstrate the effectiveness of our method rather than achieving state-of-the-art performance metrics. For instance, when we apply our method to IDDPM with a conditional setting and replace the original chain with DEIS, we observe improvements in FID. With 10 NFEs, the FID decreases from 6.65 to 6.59, and with 14 NFEs, the FID decreases from 3.67 to 3.60. It is worth noting that the symbol "†" indicates that the actual NFE is smaller than the NFE reported in the table provided by Lu et al. [44]. In summary, our method seamlessly integrates with training-free fast sampling algorithms and enables us to achieve better overall performance.

## 6.3 Ablation Study

We conduct ablation studies in Table 3 to assess the impact of different techniques. Table 3 compares four different settings, i.e., the proposed contrastive loss with a single pair $(x_t, x_j)$, a variant of contrastive loss with all samples $\sum_{i=1,i\neq t}^{T}(x_t, x_j)$ in the chain, the proposed contrastive loss with BPTT, and naive fine-tuning without contrastive loss for the same number of epochs as the previous

Table 3: FID Comparison of Ablation Study with Different Weighting Schedules on CIFAR-10. To prove the performance of BPTT and contrastive guided technique respectively, we conduct ablation study with three different settings. "EDM Baseline" [29] is the pre-trained model, "Naive Fine-Tuning" means just train more epochs as other settings. For our techniques: "Contrastive Loss" means fine-tune the model with only contrastive loss and no BPTT, "Contrastive Loss (All Steps)" is calculating the contrastive loss with all the discretization step in contrastive sampling chain and no BPTT, "Contrastive Loss + BPTT" is our setting with best performance.

|  | Weighting Schedule | | |
| Method | Fixed | Linear | Nonlinear |
| --- | --- | --- | --- |
| EDM Baseline [29] | 2.04 | - | - |
| + Naive Fine-Tuning | 2.04 | - | - |
| + Contrastive Loss | 2.04 | 1.98 | 1.99 |
| + Contrastive Loss (All Steps) | 2.18 | 2.32 | 2.39 |
| + Contrastive Loss + BPTT | 2.00 | 1.88 | 1.88 |

Table 4: FID comparison on FFHQ 64x64.

| Method | NFE↓ | FID↓ |
| --- | --- | --- |
| EDM [29] | 79 | 2.39 |
| EDM-C++ (Ours) | **63** | 2.39 |
| EDM-C++ (Ours) | 79 | **2.07** |

Table 5: FID comparison on CelebA 64x64.

| Method | NFE↓ | FID↓ |
| --- | --- | --- |
| DDPM++ [61] | 131 | 2.32 |
| Soft Diffusion [16] | 300 | 1.85 |
| INDM [33] | 132 | 1.75 |
| Diffusion StyleGAN2 [71] | 1 | 1.69 |
| STDDPM [34] | 131 | 1.90 |
| STDDPM-C++ (Ours) | **100** | 1.90 |
| STDDPM-C++ (Ours) | 131 | **1.73** |

settings. We also test three weighting schedules, including two previously mentioned dynamic weighting schedules and no schedule with a fixed weight during the entire process. As we mentioned in Section 5.2, calculating the contrastive loss with all samples in the chain disregards the consistency of the entire chain, which can destabilize pre-trained DMs on generative tasks. Therefore, it is reasonable that we obtained a worse FID when fine-tuning DMs with this setting. Moreover, the two dynamic weighting schedules show equivalent performance, both of which are much better than the fixed weight. Our results demonstrate that our contrastive method trained with BPTT for updating parameters can yield the best performance.

# 7 Conclusions

In this paper, we demonstrate the effectiveness of optimizing the KL divergence between the true sampling chain and the simulated chain at each time step in reducing the discretization error associated with numerical solvers used for solving SDEs. Our theoretical analysis supports the use of our objective function as an upper bound of the KL divergence between the data distribution and the model distribution. Notably, optimizing our objective function is equivalent to minimizing the KL divergence between the true sampling chain and the simulated chain at each time step. To address this, we propose a contrastive sampling chain that leverages the derived upper bound to reduce the discretization error. Additionally, we introduce the use of backpropagation through time (BPTT) to propagate gradients in the reverse direction of the sampling chain, and we design dynamic weighting schedules to enhance the stability of the refinement process. Our empirical results demonstrate that our approach significantly improves both the sample quality and the log-likelihood, while slightly accelerating pre-trained DMs without compromising image quality.

**Limitations and broader impact:** Although our method has shown significant improvements, there remains potential for further optimization of our method. For instance, implementing the BPTT computational graph demands a significant amount of GPU memory. Additionally, obtaining analytical solution of our weighting function will undoubtedly tighten the upper bound of the KL divergence between the data distribution and the model distribution. On the other hand, the issue of inefficient sampling remains a major obstacle to the practical application of DMs. It is reasonable to expect that the sampling speed can be greatly improved by a method that effectively optimizes the discretization error. However, it is important to acknowledge that the generation of deepfake images using our method also carries the potential risk of negative misuse of this technology.

# Acknowledgments

This work was supported by the Natural Science Foundation of China under grant 61836016. Chang Xu was supported in part by the Australian Research Council under Projects DP210101859 and FT230100549. The AI training platform supporting this work were provided by High-Flyer AI (Hangzhou High-Flyer AI Fundamental Research Co., Ltd.). This work was also supported in part by the High Performance Computing Center of Central South University. This work was supported in part by the China Scholarship Council.

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

# Appendix

## A Related Works

In this paper, our focus lies in reducing the discretization error by optimizing the Kullback-Leibler (KL) divergence at each time step between the true sampling chain and the simulated chain. To this end, our method involves fine-tuning pre-trained Diffusion Models (DMs) by minimizing a combination of the contrastive loss and the score matching loss. Thus, our objective function aims to optimize the upper bound of each KL divergence between the true sampling chain and the simulated chain. In line with this design philosophy, our method can be regarded as solving the Schrödinger Bridges [39, 40, 13]. Within this context, extensive research has been conducted to enhance DMs [57, 22, 61, 15]. In [74], a diffusion normalizing flow is proposed, which jointly trains the two neural stochastic differential equations (SDEs) [64] to minimize a common cost function that quantifies the difference between the two. By comparison, the diffusion Schrödinger Bridge optimizes an entropy-regularized transport problem on path spaces, yields diffusions which generate samples from the data distribution in finite time [18, 10]. In [69], they propose to learn a generative model via entropy interpolation [24] with a Schrödinger Bridge. This however comes with a heavy training overhead since they train the model from scratch. By contrast, our method fine-tune pre-trained DMs via reducing the gap between the true sampling chain and the simulated chain.

On the other hand, certain fast sampling methods aim to essentially reduce the discretization error in order to enhance both the sampling speed and the quality of the results. In [56] the authors use a non-Markovian forward noising. The resulted algorihtm, DDIM, achieves significant acceleration than DDPMs. More recently, the authors of [44] optimize the backward Markovian process to approximate the non-Markovian forward process and get an analytic expression of optimal variance in denoising process. Concurrent to this work, Zhang and Chen [75] propose diffusion exponential integrator sampler, which leverages a semilinear structure of the learned diffusion process to reduce the discretization error. Orthogonal to those direction, our focus lies in minimizing the discretization error by optimizing the upper bound of the Kullback-Leibler (KL) divergence between the true data distribution and the model distribution.

## B Proofs

We follow the regularity assumptions in [60, 43] to prove the Theorem 1. For completeness, we list all these assumptions in this section. We use $\mathcal{C}$ to denote all continuous functions, and let $\mathcal{C}^k$ denote the family of functions with continuous $k$-th order derivatives.

**Assumptions** We make the following assumptions throughout the paper:

(i) $p(\mathbf{x}) \in \mathcal{C}^2$ and $\mathbb{E}_{\mathbf{x} \sim p} \left[ \|\mathbf{x}\|_2^2 \right] < \infty$.

(ii) $\pi(\mathbf{x}) \in \mathcal{C}^2$ and $\mathbb{E}_{\mathbf{x} \sim \pi} \left[ \|\mathbf{x}\|_2^2 \right] < \infty$.

(iii) $\forall t \in [0, T] : \boldsymbol{f}(\cdot, t) \in \mathcal{C}^1, \exists C > 0 \forall \mathbf{x} \in \mathbb{R}^D, t \in [0, T] : \|\boldsymbol{f}(\mathbf{x}, t)\|_2 \leqslant C (1 + \|\mathbf{x}\|_2)$.

(iv) $\exists C > 0, \forall \mathbf{x}, \mathbf{y} \in \mathbb{R}^D : \|\boldsymbol{f}(\mathbf{x}, t) - \boldsymbol{f}(\mathbf{y}, t)\|_2 \leqslant C \|\mathbf{x} - \mathbf{y}\|_2$.

(v) $g \in \mathcal{C} and \forall t \in [0, T], |g(t)| > 0$.

(vi) For any open bounded set $\mathcal{O}, \int_0^T \int_{\mathcal{O}} \|p_t(\mathbf{x})\|_2^2 + D g(t)^2 \|\nabla_{\mathbf{x}} p_t(\mathbf{x})\|_2^2 \ \mathrm{d}\mathbf{x}\mathrm{d}t < \infty$.

(vii) $\exists C > 0 \forall \mathbf{x} \in \mathbb{R}^D, t \in [0, T] : \|\nabla_{\mathbf{x}} \log p_t(\mathbf{x})\|_2 \leqslant C (1 + \|\mathbf{x}\|_2)$.

(viii) $\exists C > 0, \forall \mathbf{x}, \mathbf{y} \in \mathbb{R}^D : \|\nabla_{\mathbf{x}} \log p_t(\mathbf{x}) - \nabla_{\mathbf{y}} \log p_t(\mathbf{y})\|_2 \leqslant C \|\mathbf{x} - \mathbf{y}\|_2$.

(ix) $\exists C > 0 \forall \mathbf{x} \in \mathbb{R}^D, t \in [0, T] : \|\boldsymbol{s_\theta}(\mathbf{x}, t)\|_2 \leqslant C (1 + \|\mathbf{x}\|_2)$.

(x) $\exists C > 0, \forall \mathbf{x}, \mathbf{y} \in \mathbb{R}^D : \|\boldsymbol{s_\theta}(\mathbf{x}, t) - \boldsymbol{s_\theta}(\mathbf{y}, t)\|_2 \leqslant C \|\mathbf{x} - \mathbf{y}\|_2$.

(xi) Novikov's condition: $\mathbb{E} \left[ \exp \left( \frac{1}{2} \int_0^T \|\nabla_{\mathbf{x}} \log p_t(\mathbf{x}) - \boldsymbol{s_\theta}(\mathbf{x}, t)\|_2^2 \ \mathrm{d}t \right) \right] < \infty$.

(xii) $\forall t \in [0, T] \exists k > 0 : p_t(\mathbf{x}) = O \left( e^{-\|\mathbf{x}\|_2^k} \right) as \|\mathbf{x}\|_2 \to \infty$.

Below we provide all proofs for our theorems.

**Proof.** Let $\boldsymbol{\mu}$ and $\boldsymbol{\nu}$ in the Theorem 1 denote the path measures [40, 38] of SDEs in Eq. (1) and Eq. (4) with $\lambda = 1$ respectively. Due to assumptions (i) (ii) (iii) (iv) (v) (ix) and (x), both $\boldsymbol{\mu}$ and $\boldsymbol{\nu}$ are uniquely given by the corresponding SDEs. Consider a Markov kernel $K(\{x_t\}_{t \in [0,T]}, y) := \delta(z_0 = y)$. Since $x_0 \sim p_0(x_0)$, and $x_0^{\text{SDE}} \sim p_0^{\text{SDE}}(x_0^{\text{SDE}})$, we have the following result

$$\int K(\{x_t\}_{t \in [0,T]}, x) d\boldsymbol{\mu}(\{x_t\}_{t \in [0,T]}) = p_0(x_0),$$

$$\int K(\{x_t^{\text{SDE}}\}_{t \in [0,T]}, x) d\boldsymbol{\nu}(\{x_t^{\text{SDE}}\}_{t \in [0,T]}) = p_0^{\text{SDE}}(x_0^{\text{SDE}}).$$

Here the Markov kernel $K$ essentially performs marginalization of path measures to obtain "sliced" distributions at $t = 0$. We can use the data processing inequality with this Markov kernel to obtain

$$D_{\text{KL}}\left(p_0\|p_0^{\text{SDE}}\right) = D_{\text{KL}}\left(\int K\left(\{x_t\}_{t \in [0,T]}, x\right) d\boldsymbol{\mu} \| \int K\left(\{x_t^{\text{SDE}}\}_{t \in [0,T]}, x^{\text{SDE}}\right) d\boldsymbol{\nu}\right)$$
$$\leqslant D_{\text{KL}}(\boldsymbol{\mu}\|\boldsymbol{\nu}) = \int_0^T D_{\text{KL}}(p_t\|p_t^{\text{SDE}}) dp_t. \tag{16}$$

From the perspective of the Schrödinger Bridge, the KL divergence between the true data distribution and the model distribution is bounded by the integral of KL at each time step $t \in [0, T]$ between $\boldsymbol{\mu}$ and $\boldsymbol{\nu}$. On the other hand, the value of any KL is greater than or equal to zero. Hence, we enable also to obtain

$$D_{\text{KL}}\left(p_0\|p_0^{\text{SDE}}\right) \leqslant \quad D_{\text{KL}}(\boldsymbol{\mu}_t\|\boldsymbol{\nu}_t) = \int_0^t D_{\text{KL}}(p_i\|p_i^{\text{SDE}}) dp_i. \tag{17}$$

Recall that by definition $x_t \sim p_t(x_t)$ and $x_t^{\text{SDE}} \sim p_t^{\text{SDE}}(x_t^{\text{SDE}})$, and now we assume $p_t^{\text{SDE}}$ is a prior distribution. Leveraging the chain rule of KL divergences [40], we have

$$D_{\text{KL}}(\boldsymbol{\mu}_t\|\boldsymbol{\nu}_t) = D_{\text{KL}}\left(p_t\|p_t^{\text{SDE}}\right) + \mathbb{E}_{\mathbf{z} \sim p_t}\left[D_{\text{KL}}\left(\boldsymbol{\mu}(\cdot \mid \mathbf{x}_t = \mathbf{z})\|\boldsymbol{\nu}\left(\cdot \mid \mathbf{x}_t^{\text{SDE}}\right) = \mathbf{z}\right)\right]. \tag{18}$$

Under assumptions (i) (iii) (iv) (v) (vi) (vii) (viii), the SDE in Eq. (1) has a corresponding reverse-time SDE given by

$$dx = \left[\boldsymbol{F}_t x dt - \boldsymbol{G}_t \boldsymbol{G}_t^T \nabla \log p_t(x_t)\right] dt + \boldsymbol{G}_t d\omega. \tag{19}$$

Since Eq. (19) is the time reversal of Eq. (1), it induces the same path measure $\boldsymbol{\mu}$. As a result, $D_{\text{KL}}\left(\boldsymbol{\mu}(\cdot \mid \mathbf{x}_t = \mathbf{z})\|\boldsymbol{\nu}\left(\cdot \mid \mathbf{x}_t^{\text{SDE}}\right) = \mathbf{z}\right)$ can be viewed as the KL divergence between the path measures induced by the following two (reverse-time) SDEs:

$$dx = \left[\boldsymbol{F}_t x dt - \boldsymbol{G}_t \boldsymbol{G}_t^T \nabla \log p_t(x_t)\right] dt + \boldsymbol{G}_t d\omega, x_t = x,$$

$$dx = \left[\boldsymbol{F}_t x dt - \boldsymbol{G}_t \boldsymbol{G}_t^T s_\theta(x_t^{\text{SDE}})\right] dt + \boldsymbol{G}_t d\omega, x_t^{\text{SDE}} = x.$$

The KL divergence between two SDEs with shared diffusion coefficients and starting points exists under assumptions (vii) (viii) (ix) (x) (xi) (see in [41, 66]), and can be computed via the Girsanov theorem [48]

$$D_{\text{KL}}\left(\boldsymbol{\mu}_t(\cdot \mid x_t = \mathbf{z})\|\boldsymbol{\nu}_t\left(\cdot \mid x_t^{\text{SDE}} = \mathbf{z}\right)\right)$$
$$= -\mathbb{E}_{\boldsymbol{\mu}_t}\left[\log \frac{d\boldsymbol{\nu}_t}{d\boldsymbol{\mu}_t}\right]$$
$$\overset{(j)}{=} \mathbb{E}_{\boldsymbol{\mu}}\left[\int_0^t g(i)\left(\nabla_{\mathbf{x}} \log p_i(\mathbf{x}) - s_{\boldsymbol{\theta}}(\mathbf{x}, i)\right) d\overline{\mathbf{w}}_i + \frac{1}{2}\int_0^t g(i)^2 \|\nabla_{\mathbf{x}} \log p_i(\mathbf{x}) - s_{\boldsymbol{\theta}}(\mathbf{x}, i)\|_2^2 \, di\right]$$
$$\overset{(jj)}{=} \mathbb{E}_{\boldsymbol{\mu}}\left[\frac{1}{2}\int_0^t g(i)^2 \|\nabla_{\mathbf{x}} \log p_i(\mathbf{x}) - s_{\boldsymbol{\theta}}(\mathbf{x}, i)\|_2^2 \, di\right]$$
$$= \frac{1}{2}\int_0^t \mathbb{E}_{p_i(\mathbf{x})}\left[g(i)^2 \|\nabla_{\mathbf{x}} \log p_i(\mathbf{x}) - s_{\boldsymbol{\theta}}(\mathbf{x}, i)\|_2^2\right] di$$
$$\overset{(jjj)}{\leq} \frac{1}{2}\int_0^T \mathbb{E}_{p_i(\mathbf{x})}\left[g(i)^2 \|\nabla_{\mathbf{x}} \log p_i(\mathbf{x}) - s_{\boldsymbol{\theta}}(\mathbf{x}, i)\|_2^2\right] di$$
$$= \mathcal{J}_{\text{SM}}\left(\boldsymbol{\theta}; g(\cdot)^2\right), \tag{20}$$

where (i) is due to Girsanov Theorem II [[48], Theorem 8.6.6], and (ii) is due to the martingale property of Itô integrals. Combining Eqs. (16), (18) and (20) completes the proof of Theorem 1.

Though we demonstrate that the combination of score matching losses $\mathcal{J}_{\text{SM}}\left(\boldsymbol{\theta}; g(\cdot)^2\right)$ and a KL term $D_{\text{KL}}\left(p_t \| p_t^{\text{SDE}}\right)$ is an upper bound of $D_{\text{KL}}\left(p_0 \| p_0^{\text{SDE}}\right)$, it is intractable to optimize Eq. (9) due to the unknown function form of $p_t$. To circumvent this problem, we combined with the mutual information (MI) theory [50] that MI between $p_t(x_t)$ and $p_t^{\text{SDE}}(x_t^{\text{SDE}})$ can be expressed as $I(p_t^{\text{SDE}}(x_t^{\text{SDE}}), p_j^{\text{SDE}}(x_j^{\text{SDE}})) \leq D_{KL}(p_t^{\text{SDE}} \| p_t)$, where $j \in [0, T]$ and $j \neq t$. The detailed process can be shown as

$$
\begin{aligned}
I(p_t^{\text{SDE}}(x_t^{\text{SDE}}), p_j^{\text{SDE}}(x_j^{\text{SDE}})) &\equiv \mathbb{E}_{p(x_t^{\text{SDE}}, x_j^{\text{SDE}})}\left[\log \frac{p(x_t^{\text{SDE}} \mid p_j^{\text{SDE}})}{p_t^{\text{SDE}}(x_t^{\text{SDE}})}\right] \\
&= \mathbb{E}_{p(x_t^{\text{SDE}}, x_j^{\text{SDE}})}\left[\log \frac{p(x_t^{\text{SDE}} \mid p_j^{\text{SDE}}) p_t(x_t)}{p_t(x_t) p_t^{\text{SDE}}(x_t^{\text{SDE}})}\right] \\
&= \mathbb{E}_{p(x_t^{\text{SDE}}, x_j^{\text{SDE}})}\left[\log \frac{p(x_t^{\text{SDE}} \mid p_j^{\text{SDE}})}{p_t(x_t)}\right] - D_{\text{KL}}(p_t^{\text{SDE}}(x_t^{\text{SDE}}) \| p_t(x_t)) \\
&\leq \mathbb{E}_{p_j^{\text{SDE}}(x_j^{\text{SDE}})}\left[D_{\text{KL}}(p(x_t^{\text{SDE}} \mid p_j^{\text{SDE}}) \| p_t(x_t))\right] \\
&= \mathbb{E}_{p_j^{\text{SDE}}(x_j^{\text{SDE}})}\left[D_{\text{KL}}(p_t^{\text{SDE}} \| p_t)\right].
\end{aligned}
\tag{21}
$$

Therefore, we obtain the upper bound of $I(p_t^{\text{SDE}}(x_t^{\text{SDE}}), p_j^{\text{SDE}}(x_j^{\text{SDE}}))$ given by $D_{KL}(p_t^{\text{SDE}} \| p_t)$. However, it it still intractable since the marginal distribution $p_t(x_t)$ is inaccessible. To remedy this, when applied the Jensen's inequality to this term, we enable to obtain a upper bound of $D_{KL}(p_t \| p_t^{\text{SDE}})$ by a InfoNCE loss: $I_{\text{InfoNCE}}(x_t^{\text{SDE}}, x_j^{\text{SDE}}) \geq D_{KL}(p_t \| p_t^{\text{SDE}})$ [49], where $j \in [0, T]$ and $j \neq t$. To prove this upper bound in detail, we first review the original lower bound of MI [49]

$$
I(x_t^{\text{SDE}}, x_j^{\text{SDE}}) \geq \log N - I_{\text{InfoNCE}}(x_t^{\text{SDE}}, x_j^{\text{SDE}}),
\tag{22}
$$

where $\log N$ means the total numbers of positive pair $(x_t^{\text{SDE}}, x_j^{\text{SDE}})$ when calculating the InfoNCE loss. To show clearly the connection between MI and the InfoNCE loss, we demonstrate the derivation process as follows

$$
\begin{aligned}
I_{\text{InfoNCE}}(x_t^{\text{SDE}}, x_j^{\text{SDE}}) &= -\mathbb{E}_X \log \left[\frac{\frac{p(x_t^{\text{SDE}} \mid x_j^{\text{SDE}})}{p(x_t^{\text{SDE}})}}{\frac{p(x_t^{\text{SDE}} \mid x_j^{\text{SDE}})}{p(x_t^{\text{SDE}})} + \sum_{x^- \in X_{\text{neg}}} \frac{p(x^- \mid x_j^{\text{SDE}})}{p(x^-)}}\right] \\
&= \mathbb{E}_X \log \left[1 + \frac{p(x_t^{\text{SDE}})}{p(x_t^{\text{SDE}} \mid x_j^{\text{SDE}})} \sum_{x^- \in X_{\text{neg}}} \frac{p(x^- \mid x_j^{\text{SDE}})}{p(x^-)}\right] \\
&\approx \mathbb{E}_X \log \left[1 + \frac{p(x_t^{\text{SDE}})}{p(x_t^{\text{SDE}} \mid x_j^{\text{SDE}})}(N-1) \mathbb{E}_{x^-} \frac{p(x^- \mid x_j^{\text{SDE}})}{p(x^-)}\right] \\
&= \mathbb{E}_X \log \left[1 + \frac{p(x_t^{\text{SDE}})}{p(x_t^{\text{SDE}} \mid x_j^{\text{SDE}})}(N-1)\right] \\
&\geq \mathbb{E}_X \log \left[\frac{p(x_t^{\text{SDE}})}{p(x_t^{\text{SDE}} \mid x_j^{\text{SDE}})} N\right] \\
&= -I(x_t^{\text{SDE}}, x_j^{\text{SDE}}) + \log(N).
\end{aligned}
\tag{23}
$$

Here, $x^-$ are negative instances sampled from training images. Eqs. (23) quickly becomes more accurate as $N$ increases. At the same time $\log(N) - I\left(x_t^{\text{SDE}}, x_j^{\text{SDE}}\right)$ also increases, so it's useful to use large values of $N$. In practice, we choose a bigger batch size for the large $N$. Concretely, we obtain much more samples from the given sampling chain to form the positive pairs in each training iteration. In consequence, when combined with Eq. (21), we have

$$
I_{\text{InfoNCE}}(x_t^{\text{SDE}}, x_j^{\text{SDE}}) \geq \log N - I(x_t^{\text{SDE}}, x_j^{\text{SDE}}) = \log N - D_{\text{KL}}(p_t^{\text{SDE}} \| p_t).
\tag{24}
$$

Subsequently, we apply the Jensen's inequality to further scale Eq. (24)

$$
\begin{aligned}
I_{\mathrm{InfoNCE}}(x_t^{\mathrm{SDE}}, x_j^{\mathrm{SDE}}) &\geq logN - D_{\mathrm{KL}}(p_t^{\mathrm{SDE}} \parallel p_t) \\
&= logN - p_t^{\mathrm{SDE}}(x_t^{\mathrm{SDE}}) \log \frac{p_t^{\mathrm{SDE}}(x_t^{\mathrm{SDE}})}{p_t(x_t)} \\
&\geq logN - \log \frac{p_t^{\mathrm{SDE}}(x_t^{\mathrm{SDE}}) p_t^{\mathrm{SDE}}(x_t^{\mathrm{SDE}})}{p_t(x_t)} \\
&= logN - 2 \log p_t^{\mathrm{SDE}}(x_t^{\mathrm{SDE}}) + \log p_t(x_t).
\end{aligned}
\tag{25}
$$

However, the KL in Eq. (9) is $D_{\mathrm{KL}}(p_t \parallel p_t^{\mathrm{SDE}})$ which is not equivalent to $D_{\mathrm{KL}}(p_t^{\mathrm{SDE}} \parallel p_t)$. To fill the gap, we scale the former as follows

$$
\begin{aligned}
D_{\mathrm{KL}}(p_t \parallel p_t^{\mathrm{SDE}}) &= p_t(x_t) \log \frac{p_t(x_t)}{p_t^{\mathrm{SDE}}(x_t^{\mathrm{SDE}})} \\
&\leq \log \frac{p_t(x_t) p_t(x_t)}{p_t^{\mathrm{SDE}}(x_t^{\mathrm{SDE}})} = 2 \log p_t(x_t) - \log p_t^{\mathrm{SDE}}(x_t^{\mathrm{SDE}}).
\end{aligned}
\tag{26}
$$

Therefore, when combined Eq. (25) and Eq. (26), we have

$$
\begin{aligned}
&I_{\mathrm{InfoNCE}}(x_t^{\mathrm{SDE}}, x_j^{\mathrm{SDE}}) - D_{\mathrm{KL}}(p_t \parallel p_t^{\mathrm{SDE}}) \\
&= logN - 2 \log p_t^{\mathrm{SDE}}(x_t^{\mathrm{SDE}}) + \log p_t(x_t) - [2 \log p_t(x_t) - \log p_t^{\mathrm{SDE}}(x_t^{\mathrm{SDE}})] \\
&= logN - [\log p_t^{\mathrm{SDE}}(x_t^{\mathrm{SDE}}) - \log p_t(x_t)] \\
&\stackrel{(a)}{=} logN - \log p_t^{\mathrm{SDE}}(x_t^{\mathrm{SDE}}) p_t(x_t) \\
&\geq logN,
\end{aligned}
\tag{27}
$$

where $(a)$ is due to both $p_t^{\mathrm{SDE}}(x_t^{\mathrm{SDE}}) \leq 1$ and $p_t(x_t) \leq 1$ because of the attribute of marginal distribution. Hence, it is reasonable that $p_t^{\mathrm{SDE}}(x_t^{\mathrm{SDE}}) p_t(x_t) \leq 1$ and naturally there is $\log p_t^{\mathrm{SDE}}(x_t^{\mathrm{SDE}}) p_t(x_t) \leq 0$. Accordingly, we have $I_{\mathrm{InfoNCE}}(x_t^{\mathrm{SDE}}, x_j^{\mathrm{SDE}}) \geq D_{\mathrm{KL}}(p_t \parallel p_t^{\mathrm{SDE}})$. Under such a perspective, we replace $D_{\mathrm{KL}}(p_t \parallel p_t^{\mathrm{SDE}})$ in Eq. (9) by $I_{\mathrm{InfoNCE}}(x_t^{\mathrm{SDE}}, x_j^{\mathrm{SDE}})$ and we have

$$
D_{\mathrm{KL}}\left(p_0 \parallel p_0^{\mathrm{SDE}}\right) \leq I_{\mathrm{InfoNCE}}(x_t^{\mathrm{SDE}}, x_j^{\mathrm{SDE}}) + \mathcal{J}_{\mathrm{SM}}\left(\theta; g(t)^2\right).
\tag{28}
$$

In practice, we enable to fine-tune pre-trained DMs with Eq. (28). Moreover, $\mathcal{J}_{\mathrm{SM}}\left(\theta; g(t)^2\right)$ is almost fixed in each pre-trained DM. Similarly, optimizing Eq. (28) is equivalent to minimizing $D_{\mathrm{KL}}(p_t \parallel p_t^{\mathrm{SDE}})$. Therefore, our objective function is essentially solving the Schrödinger Bridges.

## C   More Experiment Details

In this section, we will provide a more comprehensive account of our experiments, focusing on the implementation of DMs. To conduct our experiments, we utilized NVIDIA A100 GPUs for conducting experiments on CIFAR-10, CelebA, FFHQ, and ImageNet datasets. Additionally, we employed Nvidia V100 GPUs for experiments involving the combination of fast sampling methods. Based on our theoretical analysis, to optimize the KL divergence $D_{\mathrm{KL}}(p_t \parallel p_t^{\mathrm{SDE}})$, it is sufficient to add an InfoNCE loss to mitigate the discretization error. As a result, this flexible method can be applied to fine-tuning various pre-trained DMs, both with and without training-free fast sampling algorithms, thereby enhancing sample quality or achieving marginally faster sampling speeds.

To achieve this, one simply needs to include sampling chains given by pre-trained DMs or defined by fast sampling algorithms in the training process. This involves first selecting one sample $x_t^{\mathrm{SDE}}$ and then randomly choose another sample $x_j^{\mathrm{SDE}}$ from the chain to calculate the contrastive loss and form the total loss using Eq. (15). Afterwards, back-propagation through time (BPTT) is utilized to update the parameters of DMs, with the gradient flow directed in the opposite direction of this chain. It is worth noting that we maintain all the training settings of the pre-trained DMs and only modify the part that constructs the contrastive loss, thereby demonstrating the flexibility of our method. Similarly, to showcase the scalability of our method on chains defined by fast sampling methods, we replace the original chains provided by pre-trained DMs with new chains defined by fast sampling methods, while retaining all other settings the same as fine-tuning pre-trained DMs. To fine-tune a pre-trained

DM with a sampling chain of default $N$ steps in practice, we extend it to a chain of $2N$, $3N$, or even $4N$ steps to construct the contrastive loss. Once the fine-tuning process is completed, we test the performance of the $N$-step chain by drawing 50,000 samples from it and measure the widely adopted FID score, negative log-likelihood (NLL), and NFEs, where lower values indicate better performance. Ideally, a default chain in pre-trained DM with more steps will demonstrate better performance when fine-tuned by our method. However, a very long sampling chain suffers from gradient disappearance when BPTT is executed. Hence, we utilize different steps to fine-tune different DMs for different target datasets and demonstrate only the best results in this paper.

