# OpenReview forum: "Contrastive Sampling Chains in Diffusion Models"
_NeurIPS.cc/2023/Conference — NeurIPS 2023 poster_

### Official Review · Reviewer_pguc · 2023-06-29

**Soundness:** 3 good
**Presentation:** 3 good
**Contribution:** 3 good
**Rating:** 6
**Confidence:** 4

**Summary:**

This paper analysis why diffusion models need addtional contrastive loss. The main target is the reduce $D_{KL}(p_t|| p_t^{SDE})$. This paper provides a detailed theoretical analysis and solid examples to compare its method with existing ones.

**Strengths:**

1. This paper analysis the the gap $D_{KL}(p_0|| p_0^{SDE})$ to design better diffusion models. While this is not the first analysis of this object, its solution is still interesting.
2. This paper clearly analyzes and explains why contrastive loss is useful for diffusion models, as opposed to simply combining multiple losses without clear motivation. These analyses are beneficial for future research in this area.
3. The main paper's experiments incorporate the contrastive loss function along with several state-of-the-art acceleration methods, demonstrating robust performance.

**Weaknesses:**

1. In the loss function $I_{InfoNCE}(x_t^{SDE}, x_j^{SDE})$, the first item is represented by $exp(E(x_t^{SDE}), E(x_j^{SDE}))$. This item is similar to the consistent loss, which has been proven useful for diffusion models [1]. Therefore, it would be helpful if the authors could provide further analysis on the role of the second item, which is $exp(E(x_j^{SDE}), E(x^-))$, and clarify how consistent loss and contrastive loss have different influences.
2. In Section 3, this paper discusses the discretization error. However, it is not the only error present  in the sampling process. Another significant factor to consider is the estimation error arising from $s_\theta$. Although this may not influence the design of $I_{InfoNCE}(x_t^{SDE}, x_j^{SDE})$, I believe that a more detailed analysis should be provided by the authors.
3. The OOD detection results presented in Table 2 are not satisfactory. Additionally, the connection between this part and the main contribution is unclear. Are the good detection results attributed to classical diffusion models or the new loss function proposed in this paper?

[1] Daras G, Dagan Y, Dimakis A G, et al. Consistent diffusion models: Mitigating sampling drift by learning to be consistent[J]. arXiv preprint arXiv:2302.09057, 2023.

**Questions:**

From both a theoretical and experiential perspective, what are the distinct effects of consistent loss versus contrastive loss?

---

> ### Author Rebuttal · Authors · 2023-08-09
>
> ## Weaknesses
> ### 1. Provide further analysis on the role of the second item and clarify how consistent loss and contrastive loss have different influences.
> This question is quite valuable and may provide us further research direction! The contrastive loss is designed for reducing the distance between similar features and put away the dissimilar features. Hence, the contrastive loss will reduce the distance between $x_{j}^{\mathrm{SDE}}$ and $ x_{t}^{\mathrm{SDE}}$, and the second item $exp(E(x_{j}^{\mathrm{SDE}}),E(x^{\mathrm{-}}))$ of contrastive loss contribute to put away those dissimilar images $x_{j}^{\mathrm{SDE}}$ and $x^{\mathrm{-}}$. In this manner, the sampling chain will become tighter and the discretization error decreased accordingly because the distance between sampling steps decreased.
>
> By comparison, consistent loss is designed for reducing score mismatching error. In this manner, consistent loss helps DMs to generate similar images $x_{0}^{\mathrm{SDE}}$ as $x_{0}$, which enables to minish the gap between $x_{0}^{\mathrm{SDE}}$ and $x_{0}$. In a word, consistent loss is utlized during traing process to decrease the score mismatching error while our method enables to reduce the discretization error via fine-tuning pre-trained DMs.
>
> ### 2. However, it is not the only error present in the sampling process. Another significant factor to consider is the estimation error.
> The errors present in the sampling process are composed by discretization error and score mismatching error. Though those two errors both influence the final quality of generated images, our work focuses on solving discretization error caused by numerical solvers. Score mismatching error arises during training process which shows little connection with numerical solvers.
>
> ### 3. The OOD detection results presented in Table 2 are not satisfactory. Additionally, the connection between this part and the main contribution is unclear. Are the good detection results attributed to classical diffusion models or the new loss function proposed in this paper?
> Our work aims to improve generative performance of diffusion models and OOD detection is not one of our contributions. We apologize for not comprehending this question. Could you please explain it in detail?
>
>
> ## Questions
> ### 1. From both a theoretical and experiential perspective, what are the distinct effects of consistent loss versus contrastive loss?
> In a theoretical perspective, our contrastive loss is mainly focus on discretization error which caused by numerical solvers during sampling process. By contrast, the consistent loss concentrates on reducing score mismatching error during training process. On the other hand, our contrastive loss enables to combine with both deterministic sampling and stochastic sampling while consistent loss works seamless with deterministic sampling.
>
> In an experiential perspective, our contrastive loss decreases the FID on CIFAR-10 from 2.04 (random seed) to 1.88, which demonstrates a significant improvement on the pre-trained EDM. The consistent loss also helps the pre-trained EDM to reduce the FID from 1.97 (manual seed) to 1.95. Based on the above analysis, we guess consistent loss and contrastive loss can be combined to further improve DMs.

---

> > ### Comment · Reviewer_pguc · 2023-08-20
> > **Rebuttal acknowledgement**
> >
> > I have read all the reviews and author responses, and I thank the authors for their efforts. Therefore, I argee to accept this paper and keep my score.
> >
> > If OOD detection is not pertinent to your primary contribution, it may be best to remove it.

---

> > > ### Author Response · Authors · 2023-08-20
> > >
> > > We express our gratitude for your participation in the discussion and supporting our paper!
> > >
> > > We did not include the OOD detection in our paper as our sole focus was on improving pre-trained diffusion models. In my humble opinion, we are unsure whether this issue stems from our paper or other papers that you have reviewed.
> > >
> > > Once again, we express our gratitude for your thoughtful discussions, which have greatly elevated the quality of our paper!

---

### Official Review · Reviewer_AaWg · 2023-07-03

**Soundness:** 3 good
**Presentation:** 3 good
**Contribution:** 3 good
**Rating:** 7
**Confidence:** 4

**Summary:**

This paper proposes to fine-tune pre-trained diffusion models using contrastive losses to reduce discretization errors. The positive pair is formed by the same image at different steps, while the negative pair is formed by different images. To better optimize the contrastive loss, dynamic weighting schedules and back-propagation through time techniques are used.

**Strengths:**

1. The paper is well-motivated to handle an inherent problem of diffusion models, i.e., discretization errors. The theoretical conclusion is meaningful and well-aligned with the motivation.

2. The proposed method of contrastive fine-tuning with weighting schedules and BPTT is interesting and well-aligned with motivation. One advantage of this method is that it is flexible to fine-tune various off-the-shelf pre-trained diffusion models, which makes it a good contribution to the field. And it is also compatible with fast-sampling methods.

3. The experimental results are good in general, showing the effectiveness of the proposed method when combined with different baselines.

**Weaknesses:**

1. The term “Contrastive Sampling Chain” is a bit confusing to me, since the proposed method only fine-tunes the diffusion model without changing the sampling process. I would recommend using something like “Contrastive Diffusion Chain”. Likewise, the authors should avoid the saying “refinement of the sampling chain”, such as in L258-259.

2. No generated examples are given for qualitative comparisons.

3. It is unclear whether the code will be released or not.

4. In L134, the analysis is based on ODE (lambda = 0). Can this analysis be applied similarly to cases where lambda is not zero?

5. Minors issues:
- In L27, “learn” should be “learns”
- In L55, “aims” should be “aim”
- In L91, “slightly equivalent”
- In L104, “There” should be “Three”
- In L106, “diffusion” should be “diffuse”
- In L112, “modeling” should be “model”
- You should have punctuation at the end of each equation
- I would recommend adding short conclusions of the theory at the start of section 3 and section 4, for TL;DR purposes.
- The math symbols are inconsistently in bold, such as x, s, and theta.
- In L144-145, grammar errors.
- In L211, the wrong position of citations.
- In L263, should “j” be “t”?
- In reference, wrong capitalization like “gans” and “Dpm”.

**Questions:**

Please address the issues in the Weaknesses.

**Limitations:**

The limitations are discussed. I would recommend adding a discussion about the fine-tuning cost of the proposed method.

---

> ### Author Rebuttal · Authors · 2023-08-09
>
> ## Weaknesses
> ### 1. The term "Contrastive Sampling Chain" is a bit confusing to me, since the proposed method only fine-tunes the diffusion model without changing the sampling process. I would recommend using something like "Contrastive Diffusion Chain". Likewise, the authors should avoid the saying "refinement of the sampling chain", such as in L258-259.
> Thanks for your sincere review and valuable comments! We will change the term "Contrastive Sampling Chain" into a clearer presentation, where your recommendation "Contrastive Diffusion Chain" is one of our considerations. Moreover, we will fix the typos in L258-259.
>
> ### 2. No generated examples are given for qualitative comparisons.
> To qualitatively evaluate generated images, we visualize some generated images in the one-page PDF. These generated images contain rich and coherent semantic information. It is sufficient to prove the effectiveness of our method.
>
> ### 3. It is unclear whether the code will be released or not.
> We send our code to Area Chair as rebuttal policy. Moreover, we will also release our code on github after this review process.
>
> ### 4. In L134, the analysis is based on ODE (lambda = 0). Can this analysis be applied similarly to cases where lambda is not zero?
>
> Our analysis on ODE ($ \lambda =0$) is a special case of SDE ($\lambda \ne 0$). The discretization error is caused by numerical solver which shows no connection with the term $ \lambda \boldsymbol{G}_{t} d \omega $ that throwed by $ \lambda =0 $.
>
> In other words, the gap between the approximate solution and the exact solution is not come from this term $ \lambda \boldsymbol{G}_{t} d \omega $, shown in equation (6). Concretely, the integral term of equation (6) is the factor that causes the discretization error.
>
> By comparison, $ \lambda \boldsymbol{G}_{t} d \omega $ is not an integral term which can be solved without numerical solvers. Hence, Our analysis can be applied similarly to cases where lambda is not zero. The reason we present it this way is for convenience.
>
> ### 5. Minors issues.
> We will fix all these typos in our latest paper.
>
>
> ## Limitations
> ### 1. The limitations are discussed. I would recommend adding a discussion about the fine-tuning cost of the proposed method.
> For fine-tuning pre-trained EDM, our method allows achieving remarkable performance with only 20 epochs or even fewer, where each epoch costs about 130 seconds. For sampling images, the improved pre-trained EDM takes the similar overhead as the pre-trained EDM.

---

> > ### Comment · Reviewer_AaWg · 2023-08-22
> >
> > Thanks for your feedback. I will keep my score.

---

### Official Review · Reviewer_e493 · 2023-07-04

**Soundness:** 3 good
**Presentation:** 3 good
**Contribution:** 2 fair
**Rating:** 7
**Confidence:** 3

**Summary:**

This paper focuses on diffusion models (DMs). Current DMs suffer from discretization error since it leverage the numerical solvers to solve SDEs. To overcome this issue, authors propose a a contrastive loss when optimizing  DMs.  Authors present a theoretical analysis to demonstrate that combining both the generative loss and  the contrastive loss  are reasonable.  Instead of training DMs from scratch, this work optimize their model from the pratrained model.

**Strengths:**

1) This paper firstly analyze DMs suffers from  discretization error, which sounds interesting. Then to address it, a contractive loss is introduced.

2) Authors present a detail theoretical analysis about the provided problem and method.

3) This paper is well-originated.

**Weaknesses:**

1) Why it use the pretrained model? how about the performance when optimizing it from scratch? I am not convincing about the generalization capability.

2) DMs is attracting due to the large model trained on huge dataset. However, this paper only verify this proposed on small dataset, which is not convincing. I am not sure whether it still keep the advantage of the proposed method.

3) Is it expensive to visualize the generated image？  even in the supplementary material.

4) Also there are not the corresponding codes to reproduce the reported results.

**Questions:**

The method are demonstrated with both the small data and the pretrained model, which is hard to support the generalization.

**Limitations:**

1) I recommend to train the proposed method from scratch with large dataset.

2) Showing a few images is not expensive, right?

---

> ### Author Rebuttal · Authors · 2023-08-09
>
> ## Weaknesses
> ### 1.Why it use the pretrained model? how about the performance when optimizing it from scratch? I am not convincing about the generalization capability. #
> It is traditional [1-6] to improve generative performance via fine-tuning pre-trained diffusion models (DMs). For instance, [1,4] optimize the pre-trained DMs with the help of knowledge distillation techniques. Moreover, [2] utilizes discriminator guidance to refine pre-trained EDM and achieve state-of-the-art generative performance on some datasets, while [6] proposes a consistent loss to improve pre-trained EDM and greatly improve the sampling speed of DMs. Motivated by this, we propose a plug-and-play approach to enhance different DMs. Compared to training from scratch, our method allows achieving remarkable performance with training only 20 epochs or fewer when fine-tuning pre-trained DMs.
>
> On the other hand, our method remarkably improves generative performace of various pre-trained DMs, such as EDM, LSGM, STDDPM and IDDPM. Concretely, we reduce the FID of EDM and LSGM on CIFAR-10, and decrease the FID of STDDPM and EDM on CelebA and FFHQ respectively. Morver, we also conduct experiments on some training-free fast samplers and achieve obvious improvements on ImageNet dataset, seen in Table 2. This flexibility enables us to enhance various off-the-shelf pre-trained DMs, effectively demonstrating its generalization capability.
>
> [1] T. Salimans and J. Ho. Progressive Distillation for Fast Sampling of Diffusion Models. In International Conference on Learning Representations, 2021.
>
> [2] D. Kim, Y. Kim, W. Kang, and I.-C. Moon. Refining generative process with discriminator guidance in score-based diffusion models. arXiv preprint arXiv:2211.17091, 2022.
>
> [3] Z. Zhang, Z. Zhao, and Z. Lin. Unsupervised representation learning from pre-trained diffusion probabilistic models. Advances in Neural Information Processing Systems, 35:22117-22130, 2022.
>
> [4] C. Meng, R. Rombach, R. Gao, D. Kingma, S. Ermon, J. Ho, and T. Salimans. On distillation of guided diffusion models. In Proceedings of the IEEE/CVF Conference on Computer Vision and Pattern Recognition, pages 14297-14306, 2023.
>
> [5] M. Careil, J. Verbeek, and S. Lathuilière. Few-shot Semantic Image Synthesis with Class Affinity Transfer. In Proceedings of the IEEE/CVF Conference on Computer Vision and Pattern Recognition (pp. 23611-23620), 2023.
>
> [6] Y. Song, P. Dhariwal, M. Chen, and I. Sutskever. Consistency Models. arXiv e-prints, arXiv-2303, 2023.
>
>
> ### 2. DMs is attracting due to the large model trained on huge dataset. However, this paper only verify this proposed on small dataset, which is not convincing. I am not sure whether it still keep the advantage of the proposed method. #
> The largest dataset used in recent DMs works, such as **ADM [1]**, **EDM [2]**, **Dpm-solver [3]** and **FDM [6]**, as well as other DMs **[4-5]**, is the imagenet dataset. ImageNet dataset contains more than one million images which can not be regared as a "small dataset". Beyond Imagenet, these works also tend to evaluate their methods on other moderate-sized datasetes, e.g., **FFHQ** dataset, **CelebA** dataset and **CIFAR-10** dataset. We also follow this well-establsihed experiemntal setup to evaluate our algorithm. According to Table 2 in the paper, our algorithm can acheive improve performance on the ImageNet dataset. For instance, we reduce the FID from 3.67 to 3.60 with 14 steps on ImageNet when combined with DEIS-tAB3 training-free sampler. Moreover, we also we decrease the FID (lower is better) from 24.62 to 22.65 with 12 steps on ImageNet when combined with DPM-Solver-3 training-free sampler. Based on this evidence, our method is enable to keep the advantage on large dataset.
>
> [1] P. Dhariwal and A. Nichol. Diffusion models beat gans on image synthesis. Advances in Neural Information Processing Systems, 34:8780-8794, 2021.
>
> [2] T. Karras, M. Aittala, T. Aila, and S. Laine. Elucidating the design space of diffusion-based generative models. Advances in Neural Information Processing Systems 35: 26565-26577, 2022.
>
> [3] C. Lu, Y. Zhou, F. Bao, J. Chen, C. Li, and J. Zhu. Dpm-solver: A fast ode solver for diffusion probabilistic model sampling in around 10 steps. arXiv preprint arXiv:2206.00927, 2022.
>
> [4]  C. Lu, K. Zheng, F. Bao, J. Chen, C. Li, and J. Zhu. Maximum likelihood training for score-based diffusion odes by high order denoising score matching. In International Conference on Machine Learning, pages 14429-14460. PMLR, 2022.
>
> [5] S. Wizadwongsa and S. Suwajanakorn. Accelerating Guided Diffusion Sampling with Splitting Numerical Methods. In International Conference on Learning Representations, 2023.
>
> [6] W. Du, H. Zhang, T. Yang, and Y. Du. A Flexible Diffusion Model. In International Conference on Machine Learning, pages 8678-8696. PMLR, 2023.
>
> ### 3. Visualize the generated image
> To qualitatively evaluate generated images, we randomly visualize some generated images in the one-page PDF. These generated images contain rich and coherent semantic information. It is sufficient to prove the effectiveness of our method.
>
> ### 4. Also there are not the corresponding codes to reproduce the reported results.
> We send our code to Area Chair according to rebuttal policy.

---

> > ### Comment · Reviewer_e493 · 2023-08-20
> > **Convincing rebuttal**
> >
> > Thank you, authors, for the rebuttal. Your responses are quite convincing. I can see that you have made great efforts to address my concerns within the limited time frame, such as training on the Imagenet dataset. Additionally, the code has been provided. I have looke other reviewers' comments and have come to the conclusion that this paper makes a significant contribution to the community of DMs. As a result, I have decided to revise my score.

---

> ### Author Response · Authors · 2023-08-13
>
> Dear Anonymous Reviewer e493,
>
> We sincerely thank you for kindly reviewing our paper and providing valuable comments! We believe that we have fully addressed the concerns you raised. If you have any additional comments about our paper, we are more than willing to discuss them in detail and make the necessary improvements accordingly.
>
> Best,
>
> Paper2687 Authors

---

> ### Comment · Area_Chair_bkSW · 2023-08-17
> **Conf**
>
> Dear Reviewer e493,
>
> The authors of this submission just prepared a reply to concerns. Would you please check the authors' rebuttal and see whether your concerns have been addressed or not?
>
> Best regards,
> Your AC

---

> ### Comment · Area_Chair_bkSW · 2023-08-20
>
> Dear Reviewer e493,
>
> Thank you very much for your great efforts in reviewing the referred submission. I notice that you have not yet responded to the author's rebuttal. As the discussion period is about to close, would please double the authors' response and make a final decision? Your final judgment is very important to the PC to make final decisions.
>
> Best regards,
>
> Your AC

---

### Official Review · Reviewer_sJua · 2023-07-04

**Soundness:** 3 good
**Presentation:** 4 excellent
**Contribution:** 3 good
**Rating:** 8
**Confidence:** 5

**Summary:**

This paper employs the contrastive loss to construct a contrastive sampling chain, which optimizes the KL divergence between the true sampling chain and the simulated chain at each time step to reduce the discretization error associated with numerical solvers used for solving SDEs. Experimental results demonstrate that this method improves sample quality and log-likelihood, while slightly accelerating pre-trained DMs.

**Strengths:**

++ The authors provide a comprehensive error analysis, theoretical analysis, and theoretical proof regarding the causes and upper bounds of discretization error between the true distribution and its corresponding model distribution.

++ This paper conducts extensive experiments to demonstrate the performance on pre-trained diffusion models and fast samplers , and provides ablation studies to assess the impact of different techniques.

++ Reducing the discretization error is an important area for diffusion models optimization. This paper offers a novel solution by minimizing discretization error through optimizing the upper bound of the Kullback-Leibler (KL) divergence between the true sampling chain and a simulated chain at each time step.

**Weaknesses:**

-- It is preferable to provide evaluation metrics and visual results for the outputs of generative models, rather than solely focusing on the performance of enhancing pre-trained diffusion models.

-- The formatting of Tables 1, 2, and 4 could be improved, for instance, by subdividing them into several sub-tables to enhance readability, rather than separating different settings with a grey background.

**Questions:**

1. How should individual tasks choose between the linear weighting schedule and the nonlinear weighting schedule?

2. What is the expression for bate(t)?

3. Figure 1 provides limited information. It might be beneficial to incorporate additional details.

**Limitations:**

The authors refine the diffusion models by optimizing the upper bound of the KL divergence between the true sampling chain and a simulated chain. The remaining limitations and broader impact are discussed towards the conclusion of the paper.

---

> ### Author Rebuttal · Authors · 2023-08-09
>
> ## Weaknesses
> ### 1. It is preferable to provide evaluation metrics and visual results for the outputs of generative models, rather than solely focusing on the performance of enhancing pre-trained diffusion models.
> To qualitatively evaluate generated images, we visualize some generated images in the one-page PDF. These generated images contain rich and coherent semantic information. It is sufficient to prove the effectiveness of our method.
>
> ### 2. The formatting of Tables 1, 2, and 4 could be improved, for instance, by subdividing them into several sub-tables to enhance readability, rather than separating different settings with a grey background.
> Thanks for pointing out this. We will reformat those Tables in a more clarity manner and help others understand them quickly.
>
> ## Questions
> ### 1. How should individual tasks choose between the linear weighting schedule and the nonlinear weighting schedule?
> There is no limitation to choose weighting schedule for improving individual tasks. We empirically achieved similar results on various datasets when utilize those two weighting schedules. For instance, when fine-tune pre-trained EDM on CIFAR-10, we both obtain 1.88 FID.
>
> ### 2. What is the expression for bate(t)?
> Thank you for pointing out our typos! This is actually due to our carelessness in not expressing the $ \beta(t)$ correctly. We apologize for this misrepresentation. The correct form of $ \beta(t)$ is $\beta(t)=\alpha * (T-t)$ in equation (13) and $ \beta(t)=\alpha * \operatorname{PNSR}\left(x_{j}^{\mathrm{SDE}}, x_{t}^{\mathrm{SDE}}\right) $ in equation (14).
>
> ### 3. Figure 1 provides limited information. It might be beneficial to incorporate additional details.
> We incorporate more additional details into the Figure 1 and can be seen in the one-page PDF.

---

> > ### Comment · Reviewer_sJua · 2023-08-16
> >
> > Thanks for the rebuttal. I have no questions and keep my original rating.

---

### Official Review · Reviewer_mcPT · 2023-07-19

**Soundness:** 3 good
**Presentation:** 3 good
**Contribution:** 3 good
**Rating:** 5
**Confidence:** 3

**Summary:**

The paper targets the discretization error of the diffusion SDE sampling, especially when the number of sampling steps is small.
The authors propose a contrastive loss for diffusion sampling where instances on the same sampling chain are deemed positive pairs. The InfoNCE loss provides an upper bound on the KL divergence at time t.
It is shown that an appropriate combination of contrastive loss and score matching serves as an upper bound for the KL divergence between the data distribution and the model distribution.
The proposed training objective (equation 10) is a combination of the score matching loss and an InfoNCE loss and can be used to fine-tune any diffusion models' samplers.

Having introduced several two hyperparameters, beta(t) for balancing the two terms and temperature $\tau$), the authors experimented with different training strategies with ablation studies.
Experiments on CIFAR10 and ImageNet64 generation show that the sampler tuned with their proposed method outperforms the baseline diffusion model sampler.

**Strengths:**

The writing of the paper is clear. The motivation and presentation of the results are easy to follow.

The idea of introducing contrastive learning to DMs is interesting and novel to the best of my knowledge. The proposed methodology can be widely applicable to any pre-trained diffusion models as a post-fix, which can be valuable to the community.





**Weaknesses:**


### 1. Pre-trained encoder might be unfair

For all datasets, the authors use the pre-trained MoCo V2 encoder (I assume it was trained on ImageNet? Please correct me if I am wrong) to extract features in order to compute the infoNCE loss. This raises some concerns:

(1) The MoCo is pre-trained on datasets larger than CIFAR10 (and other small datasets), so the encoder has seen more data than diffusion models. Given that, it becomes less clear whether the improvement is coming from the contrastive loss itself or the encoder has seen more data. What makes it more concerning for me is that the gain from contrastive loss seems less significant on the more complicated ImageNet64 dataset. Intuitively, as the dataset gets more complicated, the baseline performance is weaker and may leave more room for improvement (correct me if I am wrong). I suggest the authors try training using encoders trained on CIFAR10 for evaluating diffusion models on CIFAR10 to make a more convincing case.

(2) For some forms of data, there is no pre-trained encoder, so the author proposed method may not be easily adapted. Of course we can train from scratch. But that introduces more computational overhead, which makes the method less appealing.

### 2. Marginal performance gap on more complicated data
On the ImageNet64 conditional experiment, the finetuned sampler shows only a marginal performance improvement over the default sampler. No standard deviation of the method is reported so I am not sure whether the improvement is statistically significant.

### 3. Concerns about training:
The infoNCE loss requires differentiable samples that are generated through multiple evaluations of the diffusion model's network, this may lead to a heavy computational overhead to differentiate through the entire computational graph. The extra computation cost is not very clear.

The proposed method seems brittle with many moving parts and hard to tune from task to task (different datasets, models, steps). Even only changing the sampling steps needs extra training. In comparison, the diffusion model's sampler can change flexibly with arbitrary sampling steps.

**Questions:**

1. Is the MoCo V2 encoder pretrained on ImageNet? Is the encoder in all the experiments the same (except for changing resolutions)?

2. How stable is the proposed method, regarding the randomness during the training process? How sensitive are the results to different hyperparameter choices?

3. For weighting function $\beta(t)$, is there any optimal/analytical solutions?

**Limitations:**

Limitations and broader impact have been discussed.

---

> ### Author Rebuttal · Authors · 2023-08-09
>
> ## Weaknesses
> ### 1. MoCo is pre-trained on ImageNet larger than CIFAR10.
> The MoCo V2 encoder utilized in this paper is pre-trained on ImageNet, and we use this encoder to fine-tune DMs in all our experiments. To further analysis the performance of our method, we conduct an ablation experiment on pre-trained EDM with the help of the MoCo V2 encoder pre-trained on CIFAR-10. Concretely, we keep the same training settings as previous experiments and fine-tune the pre-trained EDM with only 10 epochs. Empirically, we decrease the FID (lower is better) from 2.04 to 1.95, shown in the below Table. The 0.09 reduction of FID proved that our method effectively improves the generative performance of the pre-trained EDM.
> | Models     | FID$\downarrow$     | Encoder |
> | :---: | :---: | :------: |
> CIFAR-10
> |   EDM | 2.04 | - |
> |   EDM-C++ (Ours)  |   1.95   | MoCo V2 (CIFAR-10) |
> |   EDM-C++ (Ours)  |   1.88   | MoCo V2 (ImageNet) |
> IDDPM (ImageNet, 14 sampling steps)
> |   DPM-Solver-2  |   4.46   | - |
> |   DPM-Solver-2-C++ (Ours)   |   4.38   | MoCo V2 (ImageNet) |
> IDDPM (ImageNet, 20 sampling steps)
> |   DPM-Solver-2  |   3.42   | - |
> |   DPM-Solver-2-C++ (Ours)   |   3.36   | MoCo V2 (ImageNet) |
>
> In this paper, we utilize an encoder pre-trained on ImageNet to fine-tune pre-trained DMs. In this manner, our method helps the pre-trained EDM reduces the FID on CIFAR-10 from 1.95 to 1.88, seen in the above table. The 0.07 reduction of FID demonstrate that our method further improves the gerative performance. Hence, our contrastive sampling method contributes almost half of the FID reduction.
>
> On the other hand, our method increases the FID from 3.42 to 3.36 on ImageNet with 20 steps when utilizes DPM-Solver-2 training-free sampler to sample images for evaluation, shown in Table 2. By contrast, the encoder pre-trained on CIFAR-10 decreases the FID from 2.04 to 1.95 on CIFAR-10 dataset. The 0.06 reduction of FID on ImageNet is an order of magnitude as the 0.09 reduction of FID on CIFAR-10. Hence, our method presents a consistent improvement when fine-tune pre-trained DMs with the corresponding encoder.
> ### 2. For some forms of data, there is no pre-trained encoder.
> As shown in the above table, we achieve 0.09 reduction of FID when utilize the encoder pre-trained on CIFAR-10, while further obtain 0.07 reduction of FID when utilize the encoder pre-trained on ImageNet. In this paper, we all utilize the pre-trained encoder trained on ImageNet and improve the performance on four different datasets, including CIFAR-10, FFHQ, CelebA and ImageNet. Hence, arbitrary pre-trained encoder can be utilized for improving other forms of data, which demonstrate the flexibility and scalability of our method.
> ### 3.1. Only a marginal performance improvement on the ImageNet64 (conditional).
> Our method remarkably improves various DMs on CIFAR-10, FFHQ and CelebA, shown in Table 1 and 4. We also improve the generative performance on ImangeNet with very few sampling steps compared to prior DMs, see in Table 2. Moreover, our method demonstrates an consistent improvement on  class-conditioned 64 $\times$ 64 ImageNet. For intance, when compared to DPM-Solver3 sampler, the sampler DEIS-tAB3 reduces FID from 2.72 to 2.69 with 50 sampling steps, and decrease the FID from 2.84 to 2.81 with 30 sampling steps.
> ### 3.2. No standard deviation of the method is reported.
> Prior DMs have not shown the standard deviation results [1-3]. To demonstrate the stability of our approach, we conduct this experiment on the same settings for five times. Concretely, the FID results respectively are 1.8820, 1.8836, 1.8922, 1.8903, 1.8801, which fluctuate between 1.88 and 1.90.
>
> [1] T. Karras, M. Aittala, T. Aila, and S. Laine. Elucidating the design space of diffusion-based generative models. Advances in Neural Information Processing Systems 35: 26565-26577, 2022.
>
> [2] T. Salimans and J. Ho. Progressive distillation for fast sampling of diffusion models. In International Conference on Learning Representations, 2022.
>
> [3]  C. Lu, K. Zheng, F. Bao, J. Chen, C. Li, and J. Zhu. Maximum likelihood training for score-based diffusion odes by high order denoising score matching. In International Conference on Machine Learning, pages 14429-14460. PMLR, 2022.
> ### 4. The extra computation cost.
> For fine-tuning pre-trained EDM, our method allows achieving remarkable performance with only 20 epochs or even fewer, where each epoch costs about 130 seconds. For sampling images, the improved pre-trained EDM takes the similar overhead as the pre-trained EDM.
> ### 5. Changing the sampling steps needs extra training.
> Our method enables to improve pre-trained DMs with arbitrary training-free samplers. For instance, we utilize the DEIS-tAB3 sampler to construct a sampling chain and fine-tune the pre-trained IDDPM via propagating gradients along opposite direction of this chain. Hence, we only needs to fine-tune IDDPM once and subsequently report experimental results on different sampling steps.
> ## Questions
> ### 1. How stable is the proposed method, regarding the randomness during the training process? How sensitive are the results to different hyperparameter choices?
> To demonstrate the stability of our approach, we conduct experiments on the same settings with five times. Concretely, the FID results respectively are 1.8820, 1.8836, 1.8922, 1.8903, 1.8801, which fluctuate between 1.88 and 1.90. Actually, we only have one hyperparameter which is $\alpha$ used for controlling the weighting schedules $\beta (t)$. Concretely, our method achieves the similar results when $\alpha$ is within a certain range, which is insensitive to small fluctuations.
> ### 2. For weighting function beta(t), is there any optimal/analytical solutions?
> This is a valuable question! It may exists an analytical solution since we have derived an upper bound in equation (10). An appropriate weighting function $\beta(t)$ may change the inequality sign to equal sign in equation (10). We left it for future work.

---

> ### Author Response · Authors · 2023-08-13
>
> Dear Anonymous Reviewer mcPT,
>
> We sincerely thank you for kindly reviewing our paper and providing professional comments! We believe that we have fully addressed the concerns you raised. We once again express our gratitude for your valuable comments, which have significantly enhanced the quality of our paper. If you have any additional comments about our paper, we are more than willing to discuss them in detail and make the necessary improvements accordingly.
>
> Best,
>
> Paper2687 Authors

---

> > ### Comment · Reviewer_mcPT · 2023-08-13
> >
> > Thanks for the rebuttal! Most of my raised questions have been addressed. Two of my concerns still stand and they are closely connected.
> >
> > 1. Pre-trained encoder might be unfair
> >
> > I really appreciate the authors' added experiments with MoCo pretrained CIFAR-10. An improvement from 2.04 to 1.95 looks sound, although less significant than the ImageNet-pretrained case 1.88. I think the performance of the added contrastive part significantly depends on the pre-trained encoder. My guess is that if using encoders from CLIP models, the performance gain would be even higher.
> >
> > * It is difficult to distinguish which part, the extra info from pre-trained encoders, or the added contrastive sampling train, is contributing more to the performance.
> > * If we were to incorporate extra knowledge contained in pre-trained encoders, I am not sure whether the contrastive sampling train is the best way to go.
> >
> > 2. Marginal performance gap on more complicated data
> > In my humble opinion, an improvement from 2.84 to 2.81, or from 2.72 to 2.69 is significant.
> > In the ImageNet 64 case, with no extra information from the pre-trained encoder(also on ImageNet), the performance gain may be a more realistic reflection of the added contrastive sampling train. The performance gain seems to vanish as the data gets more complicated.
> > I am concerned about whether this method can improve bigger models, say text-to-image diffusion models.

---

> > > ### Author Response · Authors · 2023-08-16
> > >
> > > ### 1. Pre-trained encoder might be unfair
> > > Taking InfoNCE objective function Eq. (11) in the main paper as an example, we need to calculate the similarity between images, e.g., positive pair **($x_{t}^{SDE} $, $x_{j}^{SDE} $)**, and negative pairs **($x_{j}^{SDE} $, $x^{-} $)**. As far as we know, in the literature of contrastive learning, such as **CLIP**, **MoCo**, **SimCLR**, and **DINO**, along with **MAE**, this similarity is often caluclated in the feature space, rather than the pixel space. This implies that the encoder is an essential module when implementing the contrastive loss, othertwise directly calculating the similarity between image pixels is meaningless. Hence, encoder is a key component in our method since we motivated by enhancing pre-trained DMs with contrastive learning.
> > >
> > > To further evaluate the performance of our method, we conduct several experiments on the CIFAR-10 dataset employing the pre-trained EDM. For instance, we train the MoCo V2 from scratch and simultaneously fine-tune the pre-trained EDM. In this manner, we obtain 1.9391 FID which is better than 1.9507 FID when using the pre-trained MoCo V2 encoder, shown in the below Table. It's essential to highlight that both of them are exclusively trained on the CIFAR-10 dataset, without any supplementary information. The substantial enhancements we observe can be solely attributed to the efficacy of our method. Hence, our method contributes more to the performance instead of extra information. Moreover, our method is not limited to be used with a pre-trained encoder. It can achieve better results when the encoder is trained from scratch. Nevertheless, we advise opting for a pre-trained encoder due to its demand for much fewer computational resources. Fundamentally, the encoder is an essential module in our method, regardless of whether it has been pre-trained or not.
> > > | Models     | FID $\downarrow$     | Encoders |
> > > | :---: | :---: | :------: |
> > > |EDM|2.04|-|
> > > |Ours|1.9507|MoCo V2 (CIFAR-10, pre-trained)|
> > > |Ours|1.9391|MoCo V2 (CIFAR-10, training from scratch)|
> > > |Ours|1.8856|MoCo V2 (ImageNet, pre-trained)|
> > > |Ours|1.8831|CLIP (LAION-400M, pre-trained)|
> > > |Ours|1.8797|CLIP (LAION-2B, pre-trained)|
> > >
> > > We also utilize pre-trained CLIP encoders to improve the pre-trained EDM. To provide specific instances, we have achieved FID values of 1.8831 and 1.8797 utilizing two distinct CLIP encoders that were trained on the LAION-400M and LAION-2B datasets, respectively. Though a better encoder will further improve the performance, those FIDs are almost identical to the one we utilize the encoder trained on ImageNet. There is an upper limit to this improvement and a better encoder will not improve DMs indefinitely. Thus, while the inclusion of a more advanced encoder might have its merits, it's essential to recognize that the crux of our method hinges on the potency of contrastive loss rather than solely relying on an encoder enriched with extra information.
> > > ### 2. Marginal performance gap on more complicated data
> > > | Models     | FID $\downarrow$     | Sampling Steps |
> > > | :---: | :---: | :------: |
> > > ImageNet
> > > | DPM-Solver3| 2.72 | 50 |
> > > | DEIS-tAB3| 2.69 | 50 |
> > > | Ours| **2.67** |50|
> > > | DPM-Solver3|2.84|30|
> > > | DEIS-tAB3| 2.81 |30|
> > > | Ours| **2.75** |30|
> > > | DPM-Solver2|5.36|12|
> > > | Ours| **5.22** |12|
> > > | DPM-Solver2|7.93|12|
> > > | Ours| **7.78** |12|
> > >
> > > To evaluate our method on ImageNet dataset, we use training-free fast samplers rather default samplers to construct contrastive sampling chain to fine-tune IDDPM. In this manner, we obtain better results with the same sampling steps when using those fast samplers. Previously the DEIS-tAB3 sampler reduces the FID from 2.72 to 2.69 with 50 sampling steps when compared to DPM-Solver3 sampler. After using contrastive sampling chain constructed by DEIS-tAB3, we further reduce the FID from 2.69 to 2.67 with 50 sampling steps, seen in the above table. Moreover, DEIS-tAB3 previously reduces the FID from 2.84 to 2.81 with 30 sampling steps compared to DPM-Solver3. By contrast, we further reduce the FID from 2.81 to 2.75 with 30 sampling steps when using previous fine-tuned IDDPM. Hence, our method achieves a comparable level of promotion compared to those methods. Specifically, with 50 sampling steps, we attain a reduction of 0.02 in FID, whereas DEIS-tAB3 achieves a reduction of 0.03. When employing 30 sampling steps, we realize a FID reduction of 0.06, in contrast to the 0.03 reduction achieved by DEIS-tAB3.
> > >
> > > Importantly, our method has demonstrated significant improvements when employing fewer sampling steps. It is worth noting that achieving good results in smaller steps is the focus of recent DMs research. After using the contrastive sampling chain constructed by DPM-Solver2 sampler to fine-tune IDDPM, we reduce the FID from 7.93 to 7.78 with only 10 sampling steps, as well as reduce the FID from 5.36 to 5.22 with 12 sampling steps. This implies that our method is meaningful to achieve great improvements with few sampling steps.

---

> > > > ### Comment · Reviewer_mcPT · 2023-08-17
> > > >
> > > > I thank the authors for their elaboration on the empirical results and added experiments!
> > > >
> > > > Still, I think the performance gain is rather marginal, especially on more complicated data. IMHO, A more thorough hyperparameter tuning of the baseline or a lucky random seed may also result in an FID improvement of 0.0X. This casts doubts on the practical effectiveness of the proposed contrastive sampling.
> > > >
> > > > However, I am leaning towards acception and I will keep my score.

---

### Author Rebuttal · Authors · 2023-08-10

We incorporate more additional details into the Figure 1 of our paper and visualize some generated images for qualitative analysis.

---

### Comment · Area_Chair_bkSW · 2023-08-19
**Discussions are required for the submission**

Dear all reviewers，

Thank you very much for your great efforts in reviewing the referred submission. Now the authors have provided responses regarding your concerns. Would you please read the authors response and see whether your concerns have been addressed or not. You are welcome to raise further concerns if necessary so that the authors can respond to them timely. Your great service would be very important for the community to make final decisions.

Best regards，
Your AC

---

### Decision · Program_Chairs · 2023-09-21

**Decision:**

Accept (poster)

**Comment:**

This paper employs contrastive loss to construct a contrastive sampling chain to optimize the KL divergence between the true sampling chain and the simulated chain to reduce the discretization error associated with numerical solvers used for solving SDEs. All reviewers agree that the idea of introducing contrastive learning to DMs is interesting and novel. The authors provide a comprehensive error analysis, theoretical analysis, and theoretical proof regarding the causes and upper bounds of discretization error between the true distribution and its corresponding model distribution. Experimental results demonstrate that this method improves sample quality and log-likelihood, while slightly accelerating pre-trained DMs. All the reviewers are satisfied with the authors' responses to the questions.